# Activating mutations in BRAF disrupt the hypothalamo-pituitary axis leading to hypopituitarism in mice and humans

Angelica Gualtieri[1,14], Nikolina Kyprianou[1,14], Louise C. Gregory [2,14], Maria Lillina Vignola[1,14], James G. Nicholson [1], Rachael Tan[1], Shin-ichi Inoue[3], Valeria Scagliotti[1], Pedro Casado [4], James Blackburn[1], Fernando Abollo-Jimenez[1], Eugenia Marinelli[1], Rachael E. J. Besser [2], Wolfgang Högler [5,6], I. Karen Temple[7], Justin H. Davies[8,9], Andrey Gagunashvili[10], Iain C.A.F. Robinson[11], Sally A. Camper [12], Shannon W. Davis [13], Pedro R. Cutillas [4], Evelien F. Gevers[1], Yoko Aoki[3], Mehul T. Dattani [2,14] & Carles Gaston-Massuet [1,14✉]

Germline mutations in *BRAF* and other components of the MAPK pathway are associated with the congenital syndromes collectively known as RASopathies. Here, we report the association of Septo-Optic Dysplasia (SOD) including hypopituitarism and Cardio-Facio-Cutaneous (CFC) syndrome in patients harbouring mutations in *BRAF*. Phosphoproteomic analyses demonstrate that these genetic variants are gain-of-function mutations leading to activation of the MAPK pathway. Activation of the MAPK pathway by conditional expression of the *Braf^{V600E/+}* allele, or the knock-in *Braf^{Q241R/+}* allele (corresponding to the most frequent human CFC-causing mutation, BRAF p.Q257R), leads to abnormal cell lineage determination and terminal differentiation of hormone-producing cells, causing hypopituitarism. Expression of the *Braf^{V600E/+}* allele in embryonic pituitary progenitors leads to an increased expression of cell cycle inhibitors, cell growth arrest and apoptosis, but not tumour formation. Our findings show a critical role of BRAF in hypothalamo-pituitary-axis development both in mouse and human and implicate mutations found in RASopathies as a cause of endocrine deficiencies in humans.

[1] Centre for Endocrinology, William Harvey Research Institute, Barts & the London School of Medicine and Dentistry, Queen Mary University of London, London, UK. [2] Genetics and Genomic Medicine Research and Teaching Department, UCL, Great Ormond Street Institute of Child Health, London, UK. [3] Department of Medical Genetics, Tohoku University School of Medicine, Sendai, Japan. [4] Integrative Cell Signalling and Proteomics, Centre for Haemato-Oncology, Barts Cancer Institute, Queen Mary University of London, London, UK. [5] Department of Paediatrics and Adolescent Medicine, Johannes Kepler University Linz, Linz, Austria. [6] Institute of Metabolism and Systems Research, University of Birmingham, Birmingham, UK. [7] Faculty of Medicine, University of Southampton, Southampton, UK. [8] Child Health Directorate, University of Southampton, Southampton, UK. [9] Human Development and Health, Faculty of Medicine University of Southampton and Wessex Clinical Genetics Service, Southampton, UK. [10] NIHR Biomedical Research Centre, Great Ormond Street Hospital, Children NHS Foundation Trust and UCL, London, UK. [11] The Francis Crick Institute, London, UK. [12] Department of Human Genetics, University of Michigan, Ann Arbor, MI, USA. [13] Department of Biological Sciences, University of South Carolina, Columbia, SC, USA. [14] These authors contributed equally: Angelica Gualtieri, Nikolina Kyprianou, Louise C. Gregory, Maria Lillina Vignola, Mehul T. Dattani, Carles Gaston-Massuet. ✉email: c.gaston-massuet@qmul.ac.uk

RASopathies are a class of developmental syndromes that result from germline mutations in components of the Ras–RAF–MEK–ERK/mitogen-activated protein kinase signalling pathway (ERK/MAPK pathway hereafter). RASopathies include Noonan, Costello, Leopard and cardio-facio-cutaneous (CFC) syndromes, which share considerable phenotypic similarities[1,2]. CFC is a rare autosomal-dominant disorder characterised by multiple congenital anomalies, including a characteristic facial appearance, short stature, abnormalities of ectodermal tissues (hair and skin), congenital heart defects, gastrointestinal dysmotility and intellectual disability[3]. CFC is caused by mutations in *BRAF*, *MEK1* and *MEK2*; Noonan syndrome by mutations in *PTPN11*, *SOS1*, *KRAS*, *RAF1*, *SHOCK2*, *NRAS* and occasionally *BRAF* and *MEK1*; and Costello syndrome by mutations in *HRAS*[4–12]. The majority of individuals with CFC (50–75%) have heterozygous activating mutations in the ERK/MAPK effector protein kinase *BRAF*[1]. The ERK/MAPK pathway signalling results from activation of the receptor-linked tyrosine kinases by growth factors, hormones and cytokines, which then trigger an intracellular phosphorylation cascade in which Ras activates the protein kinase activity of RAF (Raf-1; A-Raf and B-Raf), which in turn phosphorylates and activates MEK1/2 leading to phosphorylation and activation of ERK1/2-MAPK. This results in different cellular events from proliferation, changes in cell differentiation, apoptosis and senescence[13]. Mutations in BRAF have a high occurrence rate in different types of tumours, including thyroid (30–50%), ovarian (30%) and colorectal cancers (5–20%), but are most predominantly found in melanomas (50–70%)[14,15]. Approximately 90% of activating BRAF mutations present in neoplasms are the result of substitution of a valine to glutamic acid at position 600: BRAF p.V600E. This mutation results in increased protein kinase activity leading to a constitutively active ERK/MAPK pathway[15]. A few studies have identified the somatic mutation BRAF p.V600E as a driver of the non-secreting benign pituitary tumour, papillary craniopharyngioma[16,17]. However, pituitary somatic mutations in BRAF pV600E have also been identified in corticotroph adenomas leading to hypersecretion of adrenocorticotropic hormone (ACTH), causing Cushing disease[18]. The differences between papillary craniopharyngioma (a non-secreting benign tumour) and ACTH-secreting adenomas with the same underlying genetic driver, BRAF p.V600E, reflect different, yet unknown, roles of oncogenic BRAF in different pituitary cell types leading to tumorigenesis.

Interestingly, RASopathies have been associated with endocrine phenotypes such as short stature due to growth hormone (GH) deficiency and pubertal delay[1,19]. However, the precise role for the ERK/MAPK pathway in the pathogenesis of endocrine deficiencies that are a component of the clinical phenotype of RASopathies has not been established.

Congenital hypopituitarism (CH) is a heterogeneous disorder with a wide range in severity and clinical presentations. It is defined by one (isolated) or more deficiencies (combined pituitary hormone deficiency (CPHD)) in the six anterior pituitary (AP) hormones, with growth hormone deficiency (GHD) being the most prevalent and often seen in isolation[20]. Septo-optic dysplasia (SOD) is a rare form of CH with a prevalence of 1:10,000 live births[21] and is often defined by the triad of hypopituitarism with subsequent endocrine deficits, midline neuroradiological defects (absent/hypoplastic corpus callosum and septum pellucidum) and optic nerve hypoplasia[22–24]. Mutations in several transcription factors or signalling molecules that control normal development of the hypothalamo-pituitary (HP) axis are associated with CH or SOD in mouse and humans[20]. However, the underlying aetiology for the majority of CH patients remains unknown.

Here, we report the association of SOD and CFC syndrome with BRAF genetic variants in five unrelated patients. Using transgenic mouse models, we show that activation of the MAPK pathway in the progenitors of the pituitary gland leads to abnormal terminal differentiation of hormone-producing cells, transient expansion of the pituitary stem cell pool followed by cell growth arrest and apoptosis leading to postnatal hypopituitarism. We demonstrate a biological role of activation of the MAPK pathway in the aetiology of pituitary hormone deficiencies, and the biological link between congenital forms of human hypopituitarism and RASopathies due to activation of the ERK/MAPK pathway. Hence, patients with RASopathies should be screened for hormone deficiencies as this could improve their comorbidities. Moreover, our findings implicate components of MAPK pathway as possible candidate genes for CH in humans.

## Results

**Identification of BRAF mutations in five patients with SOD associated with CFC syndrome.** Five patients with CFC were identified to have clinical features of SOD. The following previously reported de novo heterozygous genetic variants in *BRAF* were identified: the functionally characterised BRAF p.Q257R (patients 1 and 4)[7,10] and the partially characterised BRAF p.T241P (patient 3)[25], BRAF p.F468S (patient 2) and BRAF p.G469E (patient 5) (Fig. 1)[26,27]. All the identified mutations lead to changes in highly evolutionarily conserved amino acids (Fig. 1c). Patients from Pedigrees 1–3 were born to non-consanguineous Caucasian parents, Pedigree 4 was of consanguineous Pakistani origin, and Pedigree 5 was of non-consanguineous African origin. All had characteristic features of CFC encompassing facial dysmorphism, growth failure, feeding problems, structural cardiac abnormalities, neurodevelopmental delay and CNS abnormalities detected on magnetic resonance imaging (MRI) (clinical features are described in Supplementary Fig. 1 and Supplementary Tables 1 and 2). Due to the endocrine profile from these patients clearly showing endocrinopathies associated with brain and eye abnormalities characteristic of SOD, we reasoned that mutations in novel genes or known hypopituitarism or SOD causative genes, other than the reported *BRAF* variants, could be responsible for the observed clinical phenotype. To assess this, we performed whole-exome sequencing of the five patients. After assessing all coding and splice region variants in the genes previously associated with SOD, CH and CFC, results did not identify any potential pathogenic variants other than those in the *BRAF* gene (Supplementary Table 3). We also assessed all variants in the patients that are present in the ClinVar database as 'pathogenic' and 'likely pathogenic', and the *BRAF* variants were the only ones that could explain the disease in our patients. Together these results suggest that the clinical endocrine phenotype observed in our patients is due to *BRAF* mutations.

Patient 1 was referred at age 1.9 years for investigation of short stature (height SDS −3.6; body mass index (BMI) SDS 0.3) and recurrent hypoglycemia. GH deficiency was diagnosed at the age of 2.5 years, and GH treatment commenced at 3.6 years. Levothyroxine was commenced at 4.1 years due to a rapidly falling free $T_4$ concentration. Following the lack of pubertal onset at 14.1 years and a suboptimal response to GnRH testing (luteinizing hormone (LH) peak 4.1 IU/l), testosterone treatment was commenced. MRI revealed a small anterior pituitary and infundibulum, with midline defects.

Patient 2 was referred at the age of 0.9 years following MRI of the brain, which revealed features suggestive of SOD. She was short (height SDS −3.1), with multiple congenital abnormalities. GH and thyroid-stimulating hormone (TSH) deficiencies were diagnosed at 9.7 years. Levothyroxine was commenced at 9.7

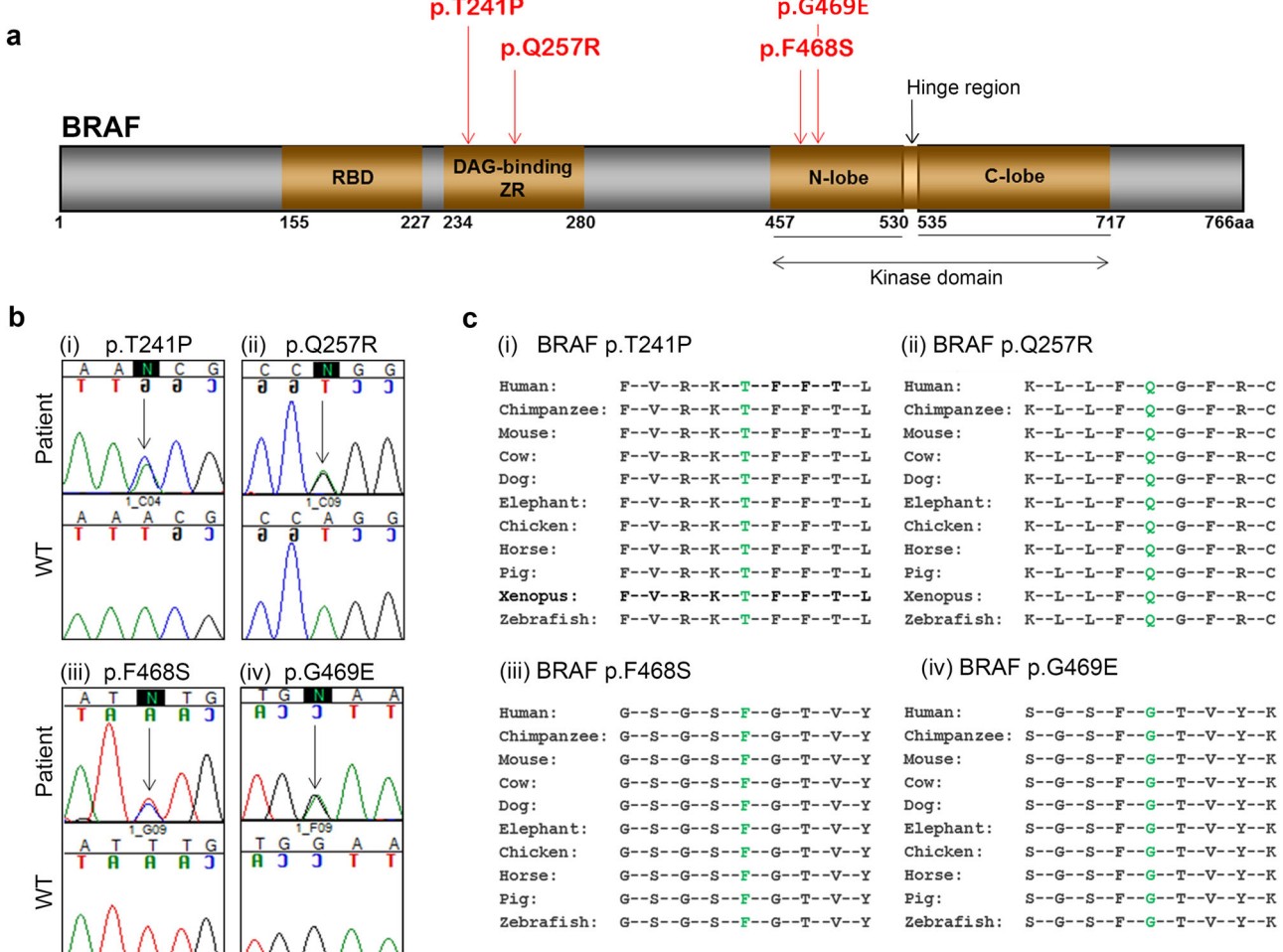

**Fig. 1 Mutations identified in hBRAF in patients with CFC and SOD. a** Schematic diagram of the hBRAF protein and the location of the mutations identified. The numbers indicate the location where each protein domain begins and ends. The mutations identified in the patients are labelled indicating the position of the substitution. **b** Electropherograms illustrating the mutations identified, indicated by an arrow and an 'N' in the sequence of each patient, with the corresponding wild-type (Wt) sequence below. (i) A heterozygous missense variant (c.721A>C) was identified in exon 6 of *BRAF* in patient 3, (ii) a heterozygous missense variant (c.770A>G) was identified in exon 6 of *BRAF* in patients 1 and 4, (iii) a heterozygous missense variant (c.1403T>C) was identified in exon 11 of *BRAF* in patient 2, (iv) a heterozygous missense variant (c.1406G>A) was identified in exon 11 of *BRAF* in patient 5. **c** Amino acid conservation of the BRAF substitutions identified in our study. (i) The threonine residue (represented by the green 'T') at position p.T241, (ii) the glutamine (represented by the green 'Q') at position p.Q257, (iii) the phenylalanine (represented by the green 'F') at position p.F468 and (iv) the glycine (represented by the green 'G') at position p.G469, and their adjacent protein sequences either side, respectively, are located at conserved regions across multiple species.

years, followed by GH at age 11.4 years. She entered puberty spontaneously at 8.3 years, but failed to progress through puberty. A GnRH test at 9.7 years demonstrated an exaggerated gonadotrophin response to GnRH stimulation. Investigations at age 13 years revealed elevated gonadotrophins (LH 44.5 IU/l, follicle-stimulating hormone (FSH) 53.5 IU/l) with an undetectable estradiol. Primary ovarian failure was diagnosed and transdermal oestrogen commenced. She subsequently died of a respiratory infection at age 16 years.

Patient 3 was referred at 5.6 years for investigation of short stature. She had a normal GH response to provocation but a low IGF1. GH was commenced and, at the age of 9 years, she had a borderline response to synacthen stimulation and hydrocortisone was started; subsequently she had a normal cortisol peak (593 nmol/l) to synacthen off hydrocortisone. She entered puberty spontaneously but failed to progress further. At 15.4 years, a GnRH test showed an exaggerated response, and she was commenced on oral oestrogen with a diagnosis of hypogonadism.

Patient 4 presented at 11.1 years with short stature. Endocrine testing revealed GH deficiency with low gonadotrophins and testosterone, and GH treatment was commenced. Despite a temporary loss to follow-up from age 13.8 to 16 years [when he had entered puberty (G3 P2 with 6 ml testes bilaterally)], he had continuously received GH treatment. GH treatment was stopped, and re-testing at 18 years (15 ml testes bilaterally) confirmed persistent GH deficiency [peak GH 3.0 µg/l, IGF1 31.8 nmol/l (NR 32.1–62.6)].

Patient 5 presented at age 3.7 years with short stature. Endocrine testing revealed normal GH secretion. The rest of the pituitary endocrinology function was normal, apart from a low IGF1. On follow-up, his growth rate is suboptimal and further investigations are planned.

**BRAF is expressed in the developing human HP axis**. The clinical phenotypes observed in our patients suggested a functional role of BRAF at the level of the forebrain and HP axis in humans. Therefore, we analysed the expression pattern of *BRAF*

during human embryonic development. *BRAF* mRNA transcripts were localised to the central nervous system, and in the developing endocrine HP axis with strong expression in the ventral diencephalon (prospective hypothalamus) and the primordium of the pituitary gland, Rathke's pouch (RP) (Supplementary Fig. 2c, d). *BRAF* expression was detected throughout the neural tube, the dorsal root ganglia, the retina and refractive lens of the developing eye, and cranial nerves. The domains of expression of *BRAF* correlate with the developmental defects observed in the patients with *BRAF* mutations, and suggest a role for mutated *BRAF* in pituitary development.

**The BRAF genetic variants are activating mutations that lead to increased kinase activity and activation of the ERK/MAPK pathway.** Previously, the BRAF variant c.770A>G (p.Q257R) was shown to result in increased ERK/MAPK pathway activity with higher levels of phosphorylated-ERK[10]. The BRAF p.T241P and the p.G469E[27,28] have been previously described in CFC patients but only partially characterised[25], indicating a mild but not statistically significant increase in phosphorylation of ERK. However, no functional studies have been performed for the p.F468S genetic variant, despite being found as a somatic mutation in sun-exposed melanoma[29] and colorectal carcinoma[30]. Therefore, to further assess the pathogenicity and functional effects of these variants on the ERK/MAPK pathway we undertook a phosphoproteomics approach using label-free mass spectrometry analyses of HEK293T cells transiently transfected with wild-type (Wt) BRAF and its variants p.T241P, p.Q257R, p.F468S and p.G469E. We used the oncogenic BRAF variant p.V600E, as a known strong activator of the ERK/MAPK pathway, and the previously characterised and most common CFC-causing mutation BRAF p.Q257R as positive controls. As expected, phosphoproteomic analyses identified increased phosphorylation of multiple components of the ERK/MAPK pathway for the oncogenic BRAF variant p.V600E and p.Q257R when compared with Wt BRAF (Fig. 2a). Interestingly, the BRAF variants p.T241P and p.F468S generated a phosphorylation pattern for the components of the ERK/MAPK pathway similar to that of BRAFp.V600E and p.Q257R (Fig. 2a). These data clearly indicate that p.T241P and p.F468S BRAF mutations also activate the ERK/MAPK pathway. Contrastingly, the BRAF p.G469E variant showed an increase in phosphorylation of proteins involved in ERK/MAPK signalling, albeit at much lower levels compared to the T241P, p.Q257R, p.F468S and p.V600E variants, suggesting a milder activation of the ERK/MAPK pathway for this variant.

Kinase substrate enrichment analysis (KSEA) showed a significantly increased kinase activity of MEK1/MEK2 and ERK1/2 for the BRAF p.V600E, p.T241P, p.Q257R and p.F468S variants when compared to Wt (Fig. 2b). In line with the peptide phosphorylation studies, KSEA estimated a milder increase in the activities of ERK1/2 and MEK1/2 for the p.G469E variant when compared to the p.T241P, p.Q257R, p.F468S and p.V600E forms (Fig. 2b). To confirm the mass spectrometry results, we assessed the levels of phosphorylated-ERK compared to those of total ERK by western blot (Fig. 2c). In agreement with the phosphoproteomic analysis, densitometry quantification of the western blot bands revealed that the p.T241P, p.Q257R, p.F468S, p.G469E and p.V600E BRAF variants led to an increased phosphorylation of ERK when compared to Wt BRAF (Fig. 2d). These data confirm that the p.T241P, p.F468S and p.G469E BRAF mutations led to activation of the ERK/MAPK pathway, although the BRAF p.G469E had a milder effect; however, it was still greater than Wt BRAF.

As expected, gene ontology analysis using the genes that encode the phosphopeptides affected by the expression of the

BRAF p.T241P, p.Q257R, p.F468S and p.G469E variants identified increased phosphorylation in proteins involved in the RAS–ERK/MAPK and the epidermal growth factor receptor (EGFR) signalling pathways (Supplementary Fig. 3). Together, our data show that the variants p.T241P, p.F468S and p.G469E result in activation of the MAPK pathways with the BRAF p.G469E having a milder activation effect compared to the p.T241P, p.Q257R and p.F468S.

**Activation of the ERK/MAPK pathway in pituitary progenitors (*Prop1:Cre;Braf^V600E/+*) results in severe postnatal hypopituitarism and lack of terminal differentiation of hormone-producing cells.** Given the hypopituitarism phenotype observed in the CFC patients, we set out to determine whether the ERK/MAPK pathway plays a role in pituitary development. We expressed the *Braf^V600E/+* allele[31] in the developing anterior pituitary gland using the *Prop1:Cre* pituitary-specific transgenic line[32]. The *Prop1:Cre* line drives expression of Cre recombinase by *Prop1* regulatory elements and efficiently expresses *Cre* in anterior pituitary[32]. However, ectopic expression of *Cre* has been reported in other tissues. To circumvent this, we crossed *Prop1:Cre* to the *Rosa26^CAGLxpSTOPLxpTomato* reporter line (*Rosa^TM/+* hereafter)[33] to obtain *Prop1:Cre;Braf^V600E/+;Rosa^TM/+*, and only embryos that exhibited Tomato expression exclusively in the pituitary gland were included in this study (Supplementary Fig. 4). Postnatally, *Prop1:Cre;Braf^V600E/+;Rosa^TM/+* (*Prop1:Cre; Braf^V600E/+* thereafter) pups showed clear signs of severe hypopituitarism with dwarfism and growth failure, and they died prematurely around weaning compared to their Wt littermates (Fig. 3a–c). Perinatal lethality was observed and after postnatal day (P) 10, only 20% of the *Prop1:Cre;Braf^V600E/+* mutants remained alive. Dissection of the pituitary glands revealed a highly hypomorphic anterior lobe (AL) consisting of only a rudimentary thin layer of cells in the mutants compared to Wt littermates (Fig. 3b, b′ and Supplementary Fig. 4i, l′). Moreover, histological sections of postnatal pituitaries revealed big cavities within the parenchyma of the AL (Supplementary Figs. 4, 6, 12, 13, 15 and 24), suggesting that tissue degeneration or death occurred in *Prop1:Cre;Braf^V600E/+* mutant pituitaries. Haematoxylin and eosin staining revealed severe morphological abnormalities with thickening of the Rathke's pouch, multiple bifurcation of the pituitary cleft and an expanded marginal zone (Supplementary Fig. 5).

To determine whether terminal differentiation of hormone-producing cells was compromised leading to the observed postnatal hypopituitarism, we examined the expression of hormones by immunohistochemistry (IHC) at E17.5 of development and postnatal day (P) 5 (Fig. 4 and Supplementary Fig. 6). Interestingly, *Prop1:Cre;Braf^V600E/+* mutant pituitaries showed complete absence of terminally differentiated somatotrophs (GH +ve); thyrotrophs (TSH+ve) and gonadotrophs (LH+ve) cells at E17.5 (Fig. 4b, d, j) with a significant increase of corticotrophs/ melanotrophs (POMC+ve cells) and lactotrophs (PRL+ve) compared to Wt (Fig. 4f, h). IHC against hormones at P5 revealed similar results, with complete absence of TSH, LH and FSH and a severely reduced number of GH+ve cells. Remarkably, at P5, the AL of the pituitary gland exhibited severe hypoplasia with a rudimentary thin layer of cells formed by mainly ACTH and PRL positive cells surrounding empty lumens and cavities (Supplementary Fig. 6). Double immunofluorescence against the pituitary stem cell marker Sox2 in combination with either POMC or PRL revealed that a large proportion of Sox2+ve cells co-express the terminal differentiation marker ACTH, and a few Sox2+ve cells also expressed PRL, which was never seen in Wt at E18.5 and P5 (Fig. 5 and Supplementary Fig. 7), suggesting that

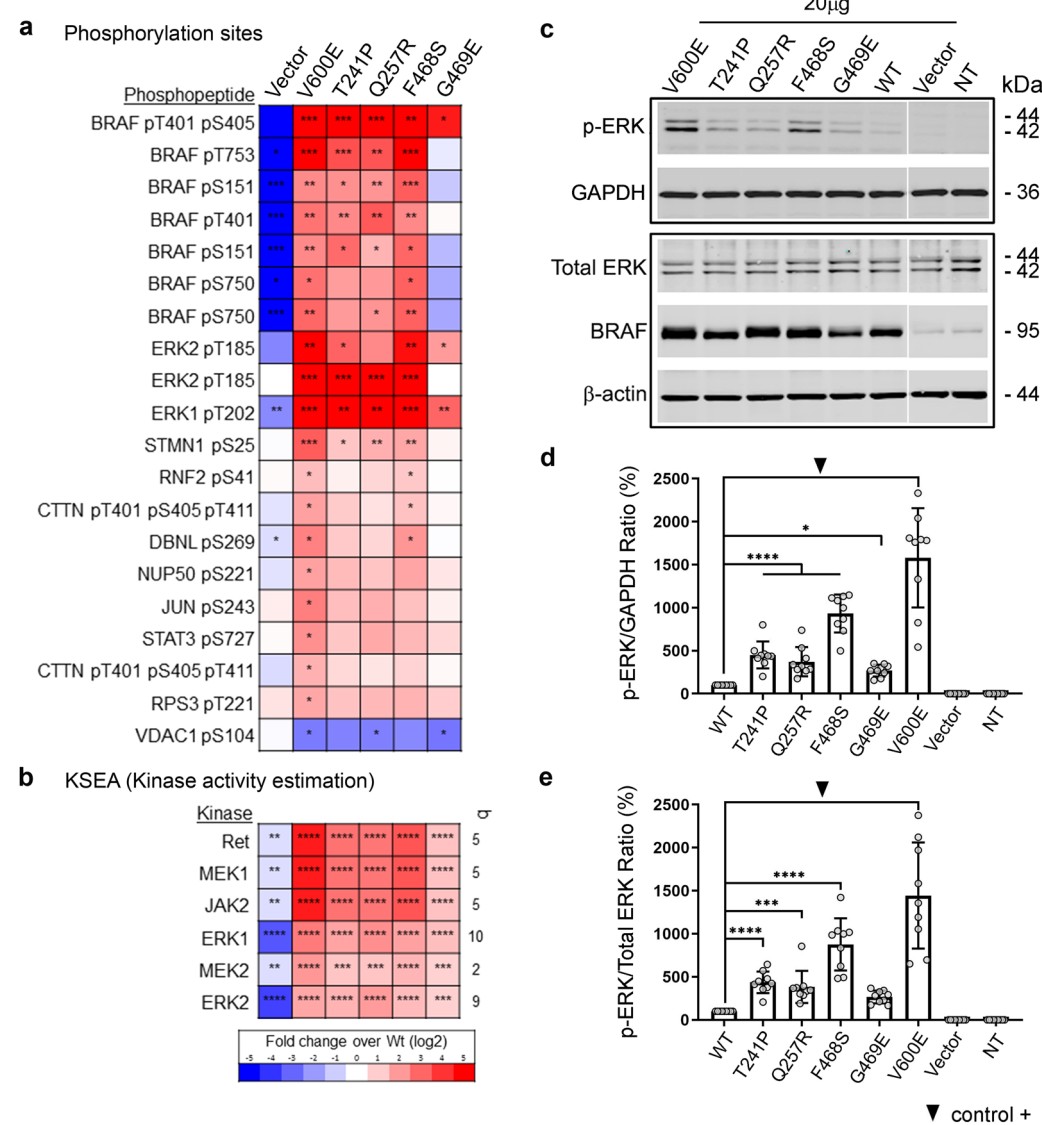

**Fig. 2 The BRAF genetic variants are pathogenic and result in activation of the ERK/MAPK pathway. a** Heat map of the phosphopeptide enrichment analyses by mass spectrometry of the BRAF variants: p.T241P, p.Q257R, p.F468S and p.G469E. These mutations result in activation of the ERK/MAPK pathway, as indicated by the increase in the ERK/MAPK phosphorylated peptides BRAF and ERK1/2. Note that the p.G469E is a mild activator with most of the peptides in blue, indicating low kinase activity. **b** KSEA for the BRAF variants p.T241P, p.Q257R, p.F468S and p.G469E compared to Wt BRAF shows an increased activity for the kinases MEK1/2 and ERK1/2 involved in the ERK/MAPK pathway, as well as an increase for JAK2 and Ret (colours represent fold change over BRAF wild-type protein expressed as Log2). **c** Western blot of cell lysates from transfected HEK293T cells with BRAF p.V600E (control) and BRAF p.T241P, p.Q257R, p.F468S and p.G469E plasmids to detect levels of total ERK and phosphorylated-ERK (p-ERK), normalised to β-actin and GAPDH. **d, e** Graphs of the western blot quantification showing increase in the p-ERK/GAPDH (**d**) and p-ERK/total ERK (**e**) ratios associated with BRAF p.T241P, p.Q257R, p.F468S and p.G469E compared to Wt BRAF (****$p < 0.0001$, ***$<0.001$ and *$<0.05$ one-way ANOVA, data represented as mean ± SD). Twenty micrograms of each BRAF variant plasmid including Wt and empty vector were used in the experiment. NT line, non-transfected control. Images are representative of nine independent experiments.

increased MAPK signalling favours Sox2+ve cells to differentiate into ACTH and PRL. Together, our results demonstrate that expression of oncogenic $Braf^{V600E}$ in developing progenitors (Prop1+ve cells) results in severe postnatal hypopituitarism due to a lack in terminal differentiation of TSH, LH and FSH and severe reduction of GH hormone-secreting cells. Interestingly, none of the postnatal pups exhibited pituitary tumours such as papillary craniopharyngioma, which is known to harbour somatic BRAF p.V600E mutations.

**The murine knock-in allele harbouring the human CFC-causing mutation BRAFp.Q257 ($CAG:Cre;Braf^{Q241R/+}$) exhibits**

**abnormalities in terminal differentiation of hormone-producing cells.** Since BRAF p.V600E is an oncogenic somatic mutation that activates the MAPK pathway but is not found in CFC patients, we studied whether the most common germline mutation identified in CFC patients (the BRAF p.Q257R), equivalent to the mouse Braf p.Q241R mutation, results in pituitary endocrine deficiencies. In this model, the $Braf^{LoxpSTOPLoxpQ241R/+}$ is ubiquitously expressed under $CAG:Cre$ reporter line[34–36]. In the C57BL/6 genetic background, this allele results in CFC-like phenotypic abnormalities, with perinatal lethality due to cardiac abnormalities as observed in CFC patients. Interestingly, $CAG:Cre;Braf^{Q241R/+}$ mutant pituitaries exhibit morphological

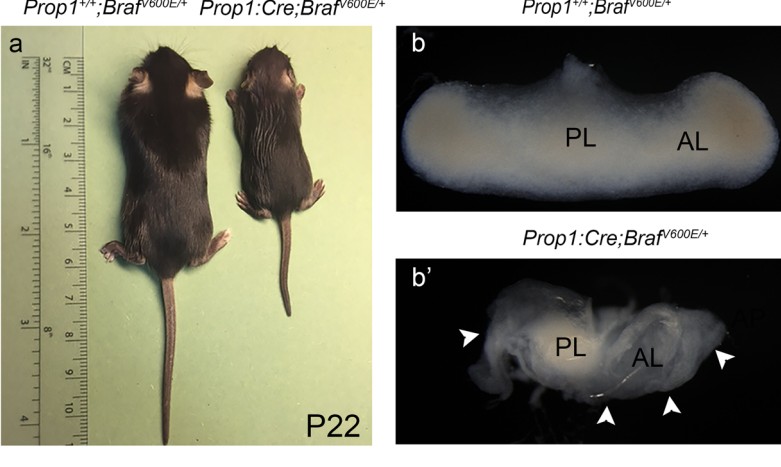

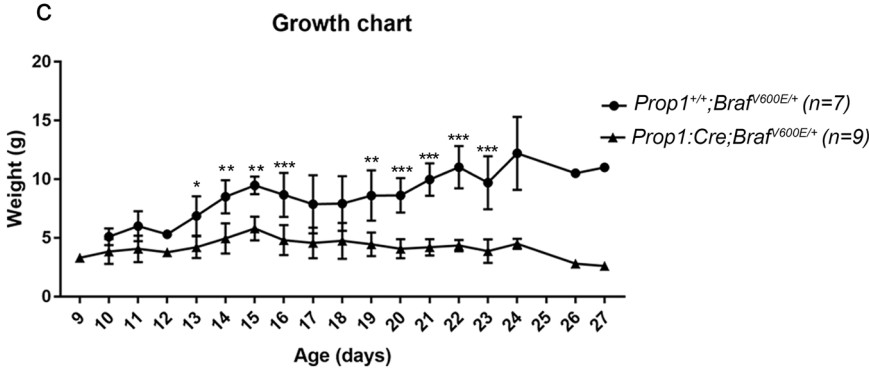

**Fig. 3 Expression of *Braf^V600E^* in the developing anterior pituitary gland (*Prop1:Cre;Braf^V600E/+^*) leads to severe hypopituitarism. a** Surviving mutant pups *Prop1:Cre;Braf^V600E/+^* exhibit dwarfism and failure to thrive compared to *Prop1^+/+^;Braf^V600E/+^* (Wt) littermates. **b**, **b'** Whole mount pictures of *Prop1^+/+^;Braf^V600E/+^* Wt (**b**) and *Prop1:Cre;Braf^V600E/+^* mutant (**b'**) pituitaries at postnatal day P22 reveal a hypoplastic anterior lobe (**b'**, AL arrowheads) composed of a rudimentary layer of cells in the mutant mice compared to Wt (**b**). **c** Growth chart illustrating growth failure of *Prop1:Cre;Braf^V600E/+^* mutants (*n* = 9) which die prematurely soon after weaning compared to Wt littermates (*n* = 7). ***$p < 0.001$; **$p < 0.01$; *$p < 0.05$ unpaired two-tailed Student's *T*-test. Data represented as mean ± SEM of *n* = 3–6 pups per genotype. AL anterior lobe, PL posterior lobe, P postnatal day.

abnormalities, consisting of pituitary cleft bifurcations and overgrowth of the marginal zone that expands into the pituitary lumen and cavities within the anterior pituitary gland (Supplementary Figs. 8 and 21). These morphological abnormalities are reminiscent of the *Prop1:Cre;Braf^V600E^* mutant pituitaries but represent a milder morphological phenotype (Supplementary Fig. 5). IHC at E18.5 revealed abnormal terminal differentiation of hormone-producing cells in these mutants, with a clear decrease in GH, TSH and LH and an increase in ACTH and PRL compared to the Wt littermates (Fig. 6). This phenotype resembles the *Prop1:Cre;Braf^V600E/+^* mutants pituitaries (Fig. 4) but with a reduced severity. Importantly, the *CAG:Cre;Braf^Q241R/+^* pituitary phenotype partially recapitulates the clinical phenotype of four of our patients. Patients 1–4, with either p.T241P, p.Q257R or p.F468S mutations, presented with GH/IGF1 deficiency, with patients 1 and 2 (harbouring p.Q257R and p.F468S respectively) also having associated TSH deficiency.

**Activation of the ERK/MAPK pathway leads to downregulation of Pit1 and Sf1 cell lineages and an increase in TPit (corticotrophs and melanotrophs).** The abnormal terminal differentiation observed in both the *Prop1:Cre;Braf^V600E/+^* and *CAG:Cre;Braf^Q241R/+^* pituitaries suggested that early cell lineage commitment transcription factors could be affected upon activation of the ERK/MAPK pathway. To ascertain this, we analysed the expression pattern of cell lineage commitment markers Pit1

(*POU1F1*) required for GH, TSH, PRL[37], Sf1 (*NR5A1*) required for LH/FSH[38], TPit/*TBX19* which gives rise to corticotroph (ACTH) and melanotroph (MSH) lineages[39,40], and the α-glycoprotein hormone subunit (α-GSU) required for gonadotrophs and Pit1-independent thyrotrophs[41]. Before performing our analyses, we tested that the onset of Cre recombinase activity from the *Prop1:Cre* transgenic line occurred prior to the appearance of the cell lineage commitment markers using the *Rosa^TM/+^* (*Rosa^CAGLxpSTOPLxpTdTomato^*) reporter from *Prop1:Cre;Braf^V600E/+^;Rosa^TM/+^* embryos. We observed positive Tomato expression from E10.5 and by E12.5, all of the RP appeared positive for Tomato, including the emergent Pomc cells (Supplementary Fig. 9). Moreover, using the *Rosa^TM/+^* allele to perform genetic lineage tracing, we identified that all the Tomato+ve cells gave rise to TPit, Pit1 and Sf1 lineages by double immunostaining against Tomato and the respective lineage commitment marker at E15.5 (Supplementary Fig. 10). These results show that Cre activity from our transgenic *Prop1:Cre* line affects all the emerging pituitary cell lineages.

Analysis of pituitary cell lineage commitment factors at E15.5 revealed that the number of TPit+ve cells was increased in the *Prop1:Cre;Braf^V600E/+^* and *CAG:Cre;Braf^Q241R/+^* embryos compared to Wt (Fig. 7a, d, g, j). The Pit1 lineage transcription factor appeared severely reduced in the *Prop1:Cre;Braf^V600E/+^* pituitaries with only a few positive cells compared to Wt (Fig. 7b, e, k); again, a consistent but milder phenotype was observed in *CAG:*

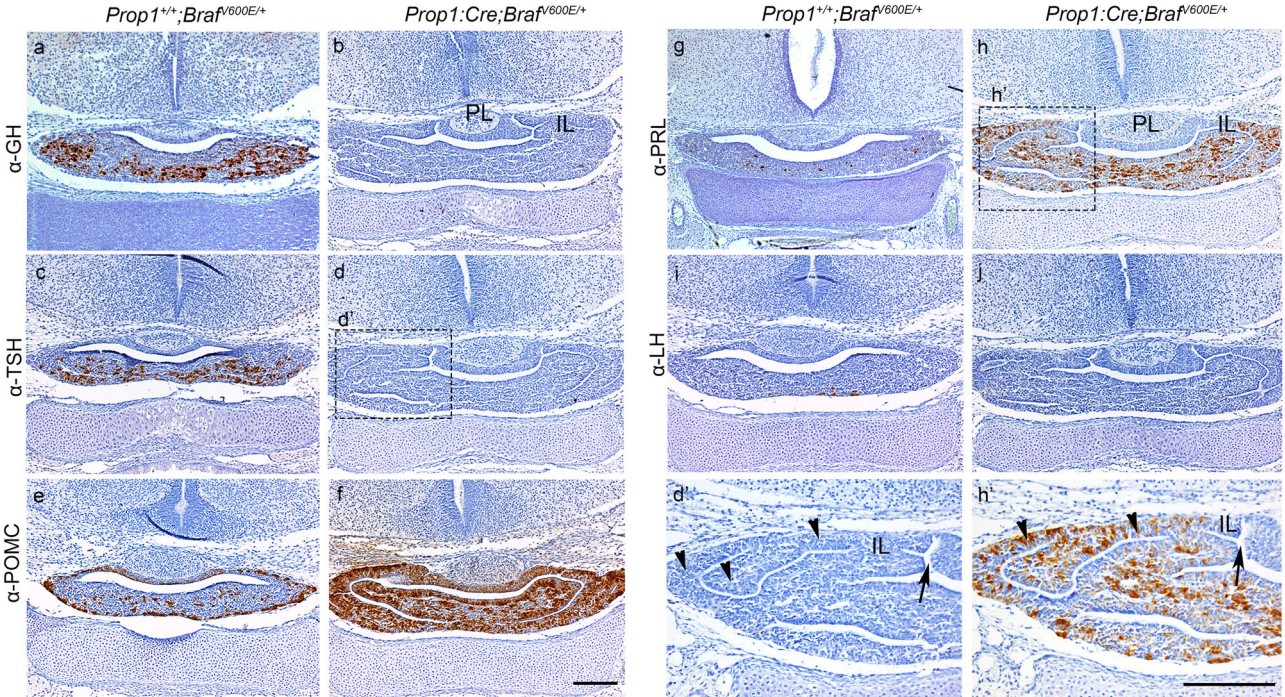

**Fig. 4 Activation of the ERK/MAPK pathway in the anterior pituitary gland (*Prop1:Cre;Braf*$^{V600E/+}$) results in defective terminal differentiation of endocrine cells. a**–**j** Immunohistochemistry against GH, TSH, POMC, PRL and LH in coronal sections through the pituitary gland of *Prop1:Cre;Braf*$^{V600E/+}$ (**b**, **d**, **f**, **h**, **j**) and Wt (**a**, **c**, **e**, **g**, **i**) embryos at E17.5 of gestation. Absence of immunoreactivity for GH, TSH, LH in *Prop1:Cre;Braf*$^{V600E/+}$ (**b**, **d**, **j**) mutant pituitaries compared to *Prop1*$^{+/+}$;*Braf*$^{V600E/+}$ (**a**, **c**, **i**) reveals deficient terminal differentiation. Note that the anterior pituitary in *Prop1:Cre;Braf*$^{V600E/+}$ is enlarged compared to Wt littermates. *Prop1:Cre;Braf*$^{V600E/+}$ pituitaries exhibit an increase in POMC (**f**) and PRL (**h**) expression compared to Wt littermates (**e**, **g**, respectively). **d′**, **h′** Higher magnification views of the squared area in **d** and **h**, respectively, revealing an expanded intermediate lobe (IL arrowheads in **d′** and **h′**) with multiple bifurcations (arrows in **d′** and **h′**). Images are representative of three embryos per genotype. IL intermediate lobe, PL posterior lobe, GH growth hormone, TSH thyroid-stimulating hormone, POMC proopiomelanocortin, PRL prolactin, LH, luteinising hormone. Scale bar: **f**, 200 μm; **h′** 500 μm.

*Cre;Braf*$^{Q241R/+}$ pituitaries (Fig. 7b, h, k). Furthermore, the gonadotroph cell lineage marker, *Sf1*, was reduced by in situ hybridisation in both *Prop1:Cre;Braf*$^{V600E/+}$ and *CAG:Cre; Braf*$^{Q241R/+}$ E15.5 pituitaries (Supplementary Fig. 11c, f, i), and similar findings were obtained on immunostaining (Supplementary Fig. 10m–r). IHC against α-GSU revealed a reduced number of α-GSU-positive cells in the *Prop1:Cre;Braf*$^{V600E/+}$ mutants compared to Wt but no evident differences were found in the *CAG:Cre;Braf*$^{Q241R/+}$ (Fig. 7c, f, i). The reduction in Pit1+ve cells was also observed in both *Prop1:Cre;Braf*$^{V600E/+}$ and *CAG:Cre; Braf*$^{Q241R/+}$ mutants at E18.5 (Supplementary Fig. 12) and postnatally at P5 in the *Prop1:Cre;Braf*$^{V600E/+}$ (Supplementary Fig. 6g, n), indicating that the downregulation of Pit1 was not due to a developmental delay. In situ hybridisation for *Pomc* and *Pit1* revealed similar results to IHC with a marked increase of *Pomc* in the *Prop1:Cre;Braf*$^{V600E/+}$ and *CAG:Cre;Braf*$^{Q241R/+}$ mutants and decreased *Pit1* mRNA expression (Supplementary Fig. 11). We then investigated whether the increase in ACTH+ve cells was due to an increased expanded domain of the melanotroph lineage marker *Pax7*[42]. No differences in the expression domain of *Pax7* were observed between Wt and the *Prop1:Cre;Braf*$^{V600E/+}$ or *CAG:Cre; Braf*$^{Q241R/+}$ mutant pituitaries (Supplementary Fig. 13).

Since we identified abnormalities in cell lineage commitment markers, we sought to determine if early pituitary specification was compromised in both *Prop1:Cre;Braf*$^{V600E/+}$ and *CAG:Cre; Braf*$^{Q241R/+}$ mutants. The expression pattern of the transcription factors implicated in early pituitary development such as *Lhx3*[43], *Prop1*[44] and *Pitx1*[45] displayed no discernible differences between mutants and Wt pituitaries at both E11.5 or E13.5 (Supplementary Fig. 14), demonstrating that activation of the

ERK/MAPK pathway does not impair the induction of Rathke's Pouch (RP).

Our data show that activation of the ERK/MAPK pathway by expressing both the *Braf*$^{V600E}$ and the *Braf*$^{Q241R}$ alleles reduces Pit1-dependent terminal differentiation of the somatotrophs (GH) and thyrotrophs (TSH), while increasing the number of ACTH+ve and PRL+ve cells. Furthermore, the TPit lineage (corticotophs and melanotrophs) was highly increased in both *Prop1:Cre;Braf*$^{V600E/+}$ and *CAG:Cre; Braf*$^{Q241R/+}$ mutant pituitaries. Together these data indicated that increased activation of the MAPK pathway affects cell lineage determination during early development of the pituitary gland.

**Activation of the MAPK pathway causes a transient increase in proliferation of the Sox2+ve progenitor cells with a decreased mitotic index at later stages of development.** The activation of the ERK/MAPK pathway has been shown to regulate proliferation in multiple systems[13,15]. We therefore measured the mitotic index (MI, % of dividing cells) in RP and AL cells at E11.5, E13.5, E15.5, E16.5, P1 and P5, using anti-phopho-histone H3 antibody (α-pHH3) by IHC (Supplementary Fig. 15). The MI was significantly increased in the *Prop1:Cre;Braf*$^{V600E/+}$ pituitaries at E11.5–13.5, but this was a transient effect, with a subsequent decrease in MI in mutant pituitaries by E16.5. This decrease in MI was exacerbated postnatally at P1 and P5 when compared with the Wt (Supplementary Fig. 15). Double immunofluorescence against pHH3 and Sox2 revealed co-localisation of these two markers, indicating that the proliferating cells are Sox2+ve pituitary progenitor/stem cells (PSCs) (Supplementary Fig. 16). Further, we used the thymidine

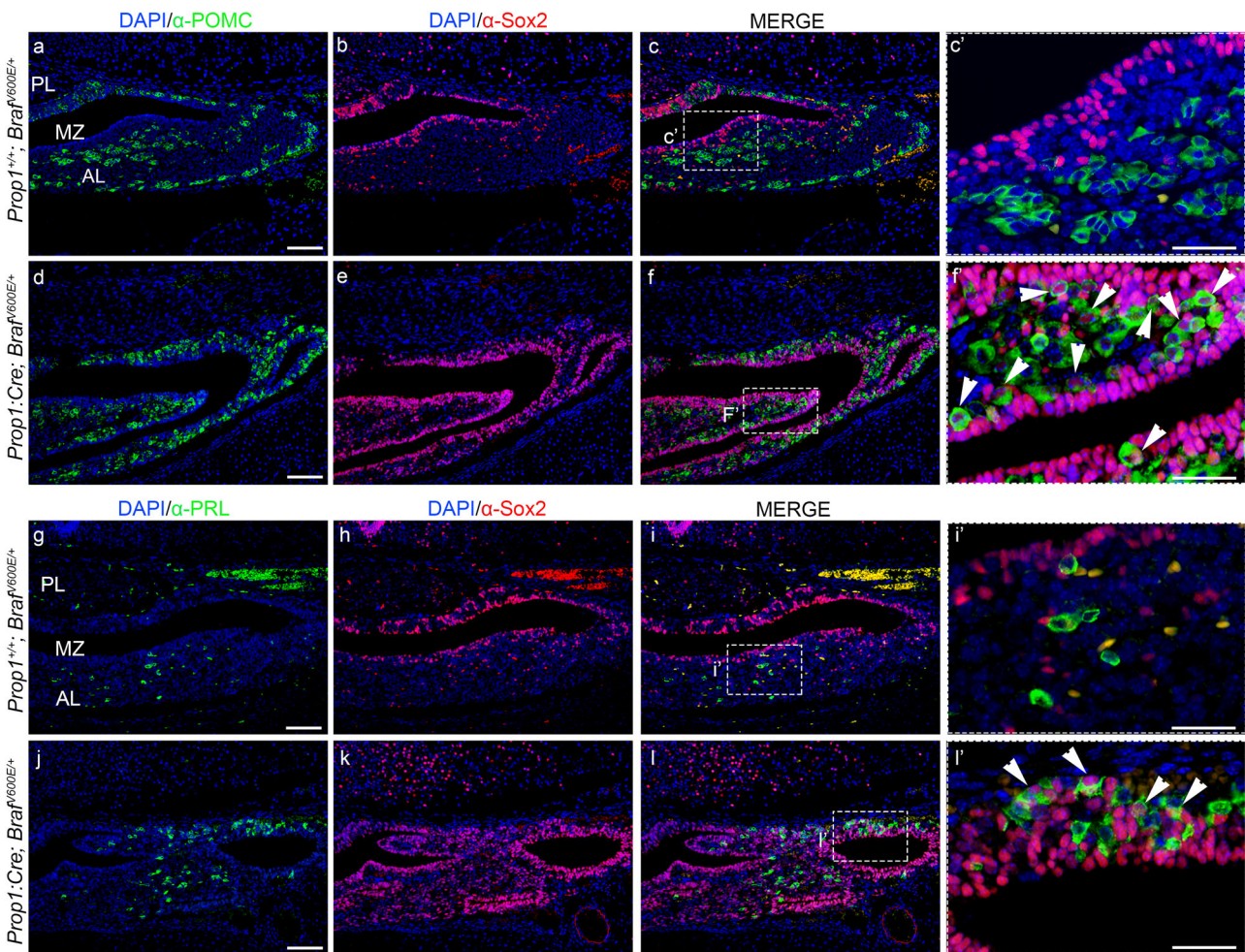

**Fig. 5 Activation of the ERK/MAPK results in increased expression of POMC and PRL with a portion of Sox2+ve progenitor/stem cells co-expressing POMC and PRL.** Double immunofluorescence (IF) against POMC (green, **a–f**), PRL (green, **g–l**) and Sox2 (red, **a–l**) on coronal sections of E18.5 Wt pituitaries (**a–c**, **g–i**) and *Prop1:Cre;Braf*$^{V600E/+}$ (**d–f**, **j–l**). The *Prop1:Cre;Braf*$^{V600E/+}$ mutant pituitaries (**d**) exhibit a higher number of POMC+ve cells compared to Wt (**a**). Enlarged merged images of the marginal zone revealed co-expression of Sox2 and POMC within a portion of POMC+ve cells (white arrowheads in **f′**). Increase in number of PRL+ve cells was observed in the *Prop1:Cre;Braf*$^{V600E/+}$ pituitaries (**j**) compared to Wt (**g**). **i′** and **l′** represent enlarged images of squared areas in **i** and **l**, respectively, showing the marginal zone. Cells expressing both Sox2 and PRL were observed in the MZ of the *Prop1:Cre;Braf*$^{V600E/+}$ mutant pituitaries (white arrowheads in **l′**), while no co-expression of Sox2 and PRL was detected in the cells of Wt pituitaries (**l′**). Images are representative of four embryos per genotype. AL anterior lobe, MZ marginal zone, PL posterior lobe. Scale bars: **a**, **d**, **g**, **j** 150 μm; **c′**, **f′**, **i′**, and **l′** 40 μm.

analogue 5-Bromo-2′-deoxyuridine (BrdU) to label cells in the S phase of the cell cycle, by treating pregnant females with a 2-h pulse of BrdU at E13.5 and E15.5. Detection of BrdU by immunofluorescence revealed that most of the BrdU+ve cells are Sox2 +ve, indicating that at this stage, most dividing cells are Sox2+ve PSCs (Supplementary Fig. 17). Moreover, quantification of percentage of BrdU cells revealed an increased in BrdU+ve incorporation at E13.5 but not at E15.5, in line with the pHH3 MI (Supplementary Fig. 15).

Since we observed a substantial increase of TPit and Pomc cells at E15.5 compared to other lineages (Fig. 7 and Supplementary Fig. 10), we asked if activation of MAPK favoured proliferation of the emerging Tpit and Pomc lineages. We performed double immunofluorescence for BrdU and TPit, Pit1 or Pomc, at both E13.5 and E15.5. At E13.5, we did not observe any co-labelling of the emerging lineage commitment markers with BrdU (Supplementary Fig. 18). At E15.5, almost no co-labelling of BrdU with cell lineage markers was observed; only very few double BrdU+ve cells co-localised with TPit or Pomc and the proportion of these

cells was similar to Wt (Supplementary Fig. 19). These experiments show that activation of MAPK does not cause overproliferation of the Tpit+ve cells, but rather leads to overproliferation of Sox2+ve undifferentiated progenitors. We then performed double immunostaining of Sox2 with either TPit, Pit1 or Pomc, which revealed a large number of Sox2+ve cells aberrantly co-expressing Tpit and Pomc (Supplementary Fig. 20). This experiment indicates that activation of MAPK favours commitment of Sox2+ve cells towards TPit- and Pomc lineages, but once these cells undergo lineage commitment, they do not over-proliferate.

Taken together, our results show that expression of Braf p. V600E results in a transient, yet severe, increase in cell proliferation of the Sox2+ve cells, resulting in an expansion of the stem cell compartment by E15.5. Additionally, increased MAPK signalling favours Sox2+ve cells to commit into Tpit- and Pomc lineages while negatively affecting Pit and Sf1 lineages. Subsequently, the proliferation rate is significantly reduced over time. We examined the proliferation in the *CAG:Cre;Braf*$^{Q241R/+}$

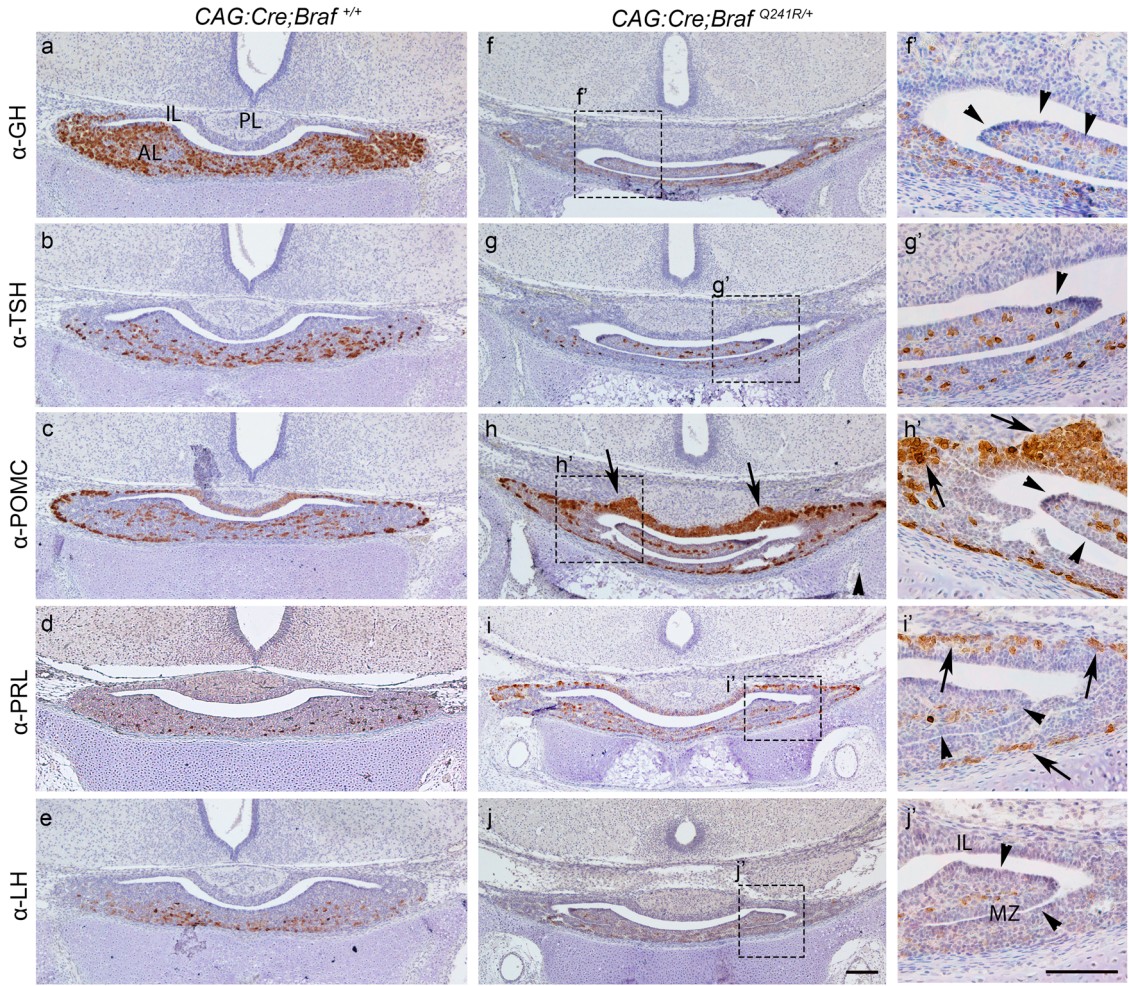

**Fig. 6 Abnormal terminal differentiation of hormone-producing cells in the *Braf*<sup>Q241R/+</sup> knock-in allele (*CAG:Cre;Braf*<sup>Q241R/+</sup>).** IHC against GH, TSH, POMC, PRL and LH hormones on coronal sections of Wt (**a**–**e**) and mutant *CAG:Cre;Braf*<sup>Q241R/+</sup> (**f**–**j**) embryos at E18.5. GH (**f**), TSH (**g**) and LH (**j**) were severely reduced in *CAG:Cre;Braf*<sup>Q241R/+</sup> mutant pituitaries compared to Wt. Increase in POMC (**h**, **h′**) and PRL (**i**, **i′**) were found in mutants compared to Wt (**c**, **d**, respectively). **f′**–**j′** represent higher magnification of the boxed areas in **f**–**j**, respectively. Note that *CAG:Cre;Braf*<sup>Q241R/+</sup> mutant pituitaries exhibit overgrowth of marginal zone (MZ) with extended growths into the pituitary lumen (arrowheads in **f′**–**j′**). Images are representative of three embryos per genotype. AL anterior lobe, IL intermediate lobe, PL posterior lobe, GH growth hormone, TSH thyroid-stimulating hormone, POMC proopiomelanocortin, PRL prolactin, LH luteinising hormone. Scale bar: **j** 200 μm; **j′** 500 μm.

pituitaries and, similar to the *Prop1:Cre;Braf*<sup>V600E/+</sup>, identified an increased MI at E13.5; however, from E15.5 no significant differences were noted (Supplementary Fig. 21).

**Braf p.V600E results in increased expression of cell senescence marker (SA)-β-galactosidase, p16<sup>INK4a</sup> and the cell cycle inhibitors p21, p27<sup>Kip1</sup> and p57<sup>Kip2</sup> leading to cell growth arrest, decreased proliferation and apoptosis of PSC in vitro.** Several reports have shown that expression of BRAF p.V600E alone is not sufficient to cause transformation and malignancy in vitro and in vivo[46–49]. Instead, BRAF p.V600E causes an initial cell proliferation, followed by growth arrest, oncogene-induced senescence (OIS) and apoptosis[50–52]. Hence, we hypothesised that expression of *Braf*<sup>V600E</sup> alone in PSC results in OIS leading to growth arrest, apoptosis and severe hypoplasia. To test our hypothesis, we performed in situ hybridisation of cell cycle inhibitors at E16.5 and P1 (Fig. 8). We chose embryonic day E16.5 as the starting developmental point because this is the stage when we first observed a decreased MI in the *Braf*<sup>V600E</sup> mutant pituitaries (Supplementary Fig. 15). Expression of the cell cycle inhibitors *Cdkn1c* (p57<sup>Kip2</sup>), *Cdkn2a* (p16<sup>INK4a</sup>), *Cdkn1a* (p21)

and *Cdkn1b* (p27<sup>Kip1</sup>) was upregulated in *Prop1:Cre;Braf*<sup>V600E/+</sup> both in E16.5 and P1 mutant pituitaries compared to Wt (Fig. 8). Quantification of mRNA using real-time quantitative reverse transcription PCR (RT-qPCR) from P1 pituitaries showed a 17.8-fold upregulation of the senescence marker *p16*<sup>INK4a</sup>, 4.6-fold upregulation of the cell cycle inhibitor *p57*<sup>Kip2</sup>, and to a lesser extent, increases in *p21* and *p27*<sup>Kip1</sup>. Double immunofluorescence of Sox2 with p57<sup>Kip2</sup> or p27<sup>Kip1</sup> confirmed abnormal co-expression and upregulation of these cell cycle inhibitors by the Sox2+ve PSCs along the marginal zone in mutant *Braf*<sup>V600E</sup> pituitaries compared to Wt (Fig. 9a–f, g–l). Moreover, immunofluorescence using the activated MAPK readout pERK revealed persistent activation of pERK in the stem cell compartment of mutant pituitaries (Supplementary Fig. 22).

To assess whether expression of Braf p.V600E causes decreased proliferation and apoptosis, we tested the effect of Braf p.V600E in PSCs isolated from E18.5, P4 and P14 pituitaries in vitro. Cultures of PSCs from *Prop1:Cre;Braf*<sup>V600E/+</sup>;*Rosa*<sup>TM/+</sup> or Wt were performed using stem cell adherent cultures, and the number of colonies and cells per colony were used as a readout of proliferative capacity (Fig. 10 and Supplementary Fig. 23). Following culture, 98% of the cells were positive for Tomato,

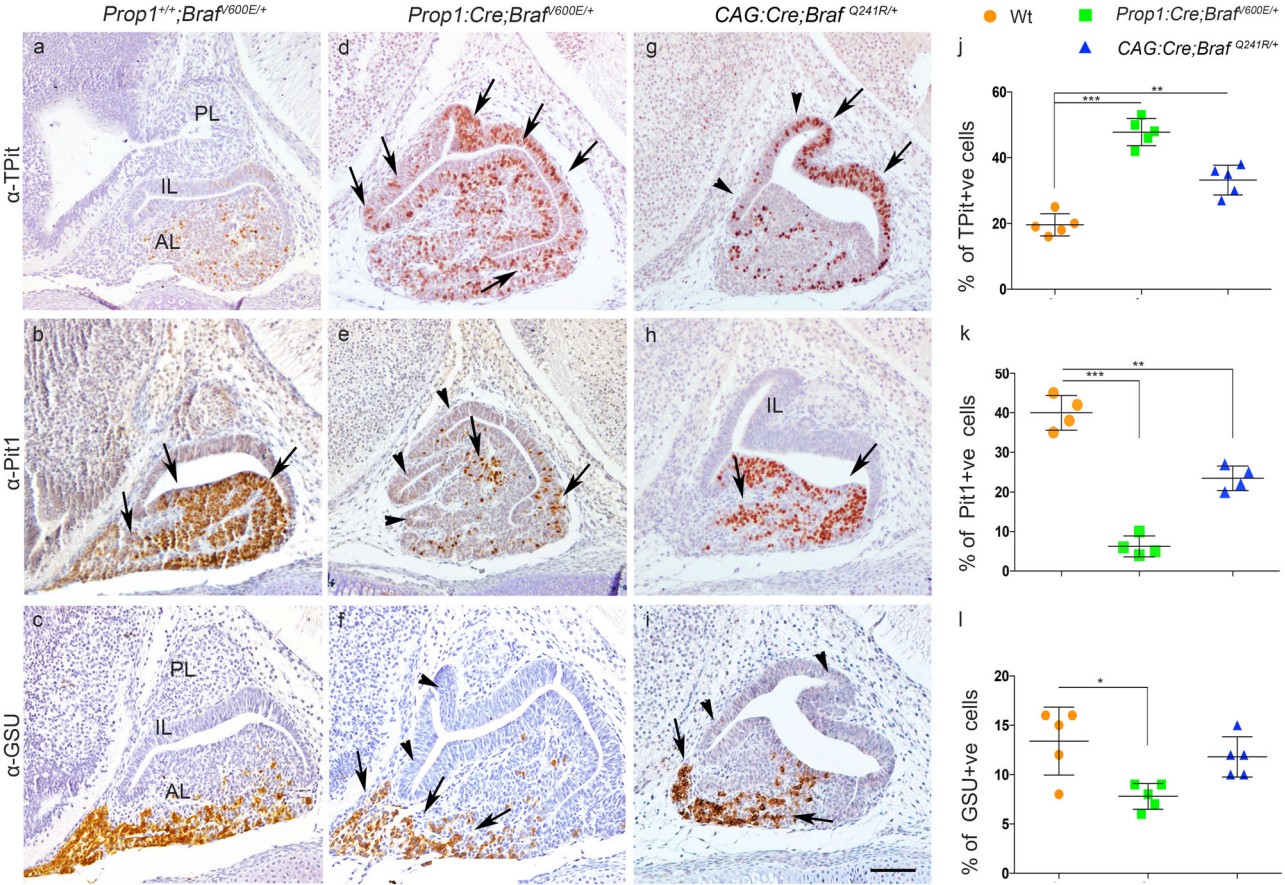

**Fig. 7 Expression of *Braf^V600E* and *Braf^Q241R* leads to abnormal cell lineage specification with increase in TPit (corticotrops and melanotrophs) and decrease in Pit1 (somatotrops, thyrotrophs and lactotrophs).** IHC against TPit (**a**, **d**, **g**), Pit1 (**b**, **e**, **h**) and α-GSU (**c**, **f**, **i**) on sagittal section of E15.5 embryos of Wt (**a–c**), *Prop1:Cre;Braf^V600E/+* (**d–f**) and *CAG:Cre;Braf^Q241R/+* (**g–i**). Expression of TPit was increased in the *Prop1:Cre;Braf^V600E/+* and *CAG:Cre;Braf^Q241R/+* pituitaries compared to Wt (arrows in **d**, **g**). Quantification of TPit-positive cells shows statistically significant increase in the % of TPit+ve cells in both *Prop1:Cre;Braf^V600E/+* and *CAG:Cre;Braf^Q241R/+* pituitaries compared to Wt (**j**). Severe reduction of Pit1 immunoreactivity was observed in *Prop1:Cre;Braf^V600E/+* with only few positive foci (arrows in **e**) compared to Wt (**b**). Quantification of the Pit1-positive cells revealed a decrease in Pit1 cells in *Prop1:Cre;Braf^V600E/+* and *CAG:Cre;Braf^Q241R/+* mutant pituitaries (**k**). Mild reduction of α-GSU was observed in *Prop1:Cre;Braf^V600E/+* pituitaries (arrows in **f**) (**l**). Note that *Prop1:Cre;Braf^V600E/+* and *CAG:Cre;Braf^Q241R/+* pituitary glands exhibited morphological abnormalities with expanded overgrowth and bifurcations of IL (arrowheads **d–f** and **g–i**) and overall enlarged size. Quantification of percentage of TPit (**j**), Pit1 (**k**) and α-GSU-positive cells (**l**) (***p < 0.001; **p < 0.01; *p < 0.05 one-way ANOVA, data represented as mean ± SEM from n = 4–5 pituitaries per genotype). AL anterior lobe, IL intermediate lobe, PL posterior lobe, IHC immunohistochemistry. Images are representative of four or five embryos per genotype. Scale bar: **i** 200 μm.

demonstrating Cre activity and recombination of the *Rosa^TM* allele. Moreover, expression of the *Braf^V600E/+* allele was assessed both by western blotting using a specific Braf p.V600E antibody and by immunofluorescence (Supplementary Fig. 23c, m). Braf p. V600E -expressing PSCs fail to show overt differences in proliferation as assessed by the number of colonies and number of cells per colony at E18.5 (Supplementary Fig. 23a, b). To demonstrate that the ERK/MAPK pathway had been activated, we performed western blot and immunofluorescence against phosphorylated-ERK, which demonstrated increased levels of phosphorylated-ERK in the *Prop1:Cre;Braf^V600E/+;Rosa^TM/+* cells compared to Wt. A significant increase in the senescence markers such as senescence-associated (SA)-β-galactosidase, p16^INK4a, p57^Kip2 and p21 was observed in mutant PSCs compared to Wt PSCs (Supplementary Fig. 23e, l; h, o; i, p; j, q). Importantly, the colony-forming capacity of the mutant PSCs at both P4 and P14 was severely compromised postnatally, when the pituitary hypoplasia is evident in vivo (Fig. 10a–d). TUNEL immunofluorescence revealed increased numbers of apoptotic cells in

mutant PSCs compared to Wt and a significantly decreased MI, with less pHH3+ve cells per colony in Braf p.V600E -expressing mutant PSCs compared to Wt (Fig. 10g–h, i, j, n–o). Taken together, our data show that expressing Braf p.V600E in PSCs leads to cell growth arrest, with increase in the expression of senescence markers p16^INK4a, p21, SA-β-galactosidase and cell cycle inhibitors, leading to a reduction in colony formation and an increased apoptosis of PSCs in vitro. Since we observe an increase in TUNEL+ve cells in PSCs, we reasoned that the hypoplastic pituitary phenotype of postnatal *Prop1:Cre;Braf^V600E/+* mutants could be due to a combination of both reduced proliferation and increased apoptosis. Therefore, we assessed apoptosis in the pituitary glands of *Prop1:Cre;Braf^V600E/+* and Wt at three stages (E16.5, P1 and P5) by using an anti-activated cleaved CASPASE antibody (Supplementary Fig. 24). Quantification of CASPASE+ve cells revealed a significant increase in apoptotic cells in the *Prop1:Cre;Braf^V600E/+* mutant pituitaries compared to their Wt littermates at E16.5, P1 and P5, indicating that expression of Braf p.V600E leads to an increase in apoptosis.

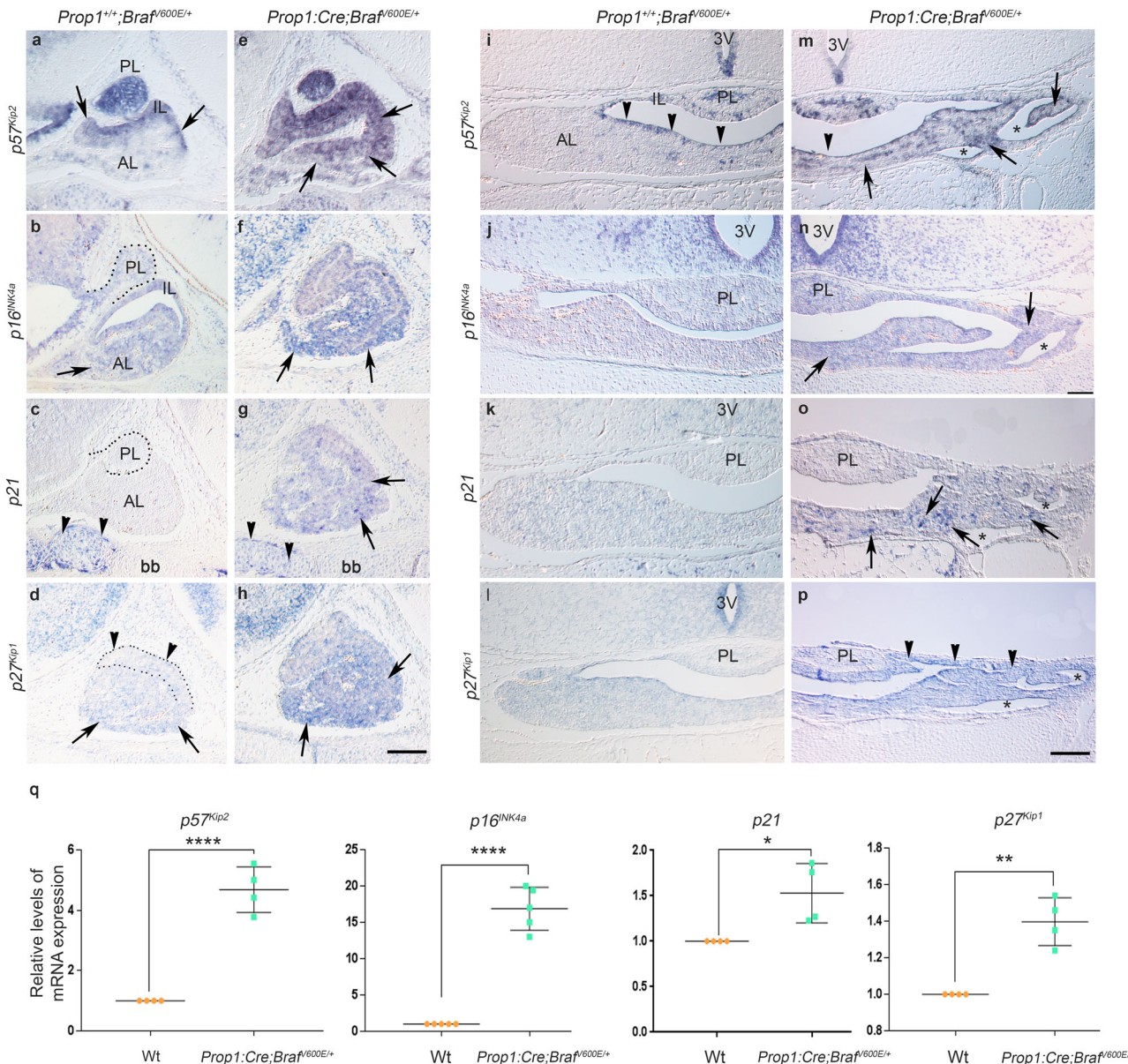

**Fig. 8 Expression of *Braf^V600E* results in upregulation of cell cycle inhibitors *p57^Kip2*, *p27^Kip1* and the senescence markers *p16^INK4a* and *p21*. a–h** In situ hybridisation of sagittal sections through embryonic pituitary gland of Wt (**a–d**) and *Prop1:Cre;Braf^V600E/+* mutant pituitaries (**e–h**) at E16.5 reveals significantly increased *p57^Kip2*, *p16^INK4a*, *p21* and *p27^Kip1* mRNA transcripts in mutant pituitaries. *p57^Kip2* transcripts were upregulated and its expression domain was expanded ventrally (arrows in **e**). *p16^INK4a* mRNA transcripts were upregulated in the ventral portion of the AL (arrows in **f**). *p21* transcripts were located in the AL in mutant pituitaries (arrows in **g**) and absent in Wt (**c**), although *p21* was expressed in the basisphenoid bone (bb, arrowheads in **c**, **g**) in Wt. Expression of *p27^Kip1* was significantly upregulated in the ventral side of the AL (arrows in **h**) compared to Wt (arrows in **d**). The IL was negative for *p27^Kip1* (arrowheads in **d**). **i–p** Representative coronal sections at P1 of Wt **i–l** and *Prop1:Cre;Braf^V600E/+* mutant pituitaries **m–p**. *p57^Kip2* mRNA transcripts were localised mainly in the IL and the MZ (arrowheads in **i**) while in the mutants expression was found ectopically throughout the AL (arrows in **m**). Expression of *p16^INK4a* (arrows in **n**), *p21* (arrows in **o**) and *p27^Kip1* (arrowheads in **p**) was upregulated compared to the corresponding Wt pituitaries (**j–l**). Images are representative of five embryos per genotype. Asterisks indicate tissue cavities within the AL. **q** Quantitative RT-qPCR from P1 pituitary glands revealed increased mRNA expression of *p57^Kip2* (4.6-fold increase), *p16^INK4a* (17.81-fold increase), *p21* and *p27^Kip1* compared to Wt (****$p < 0.0001$; **$p < 0.01$; *$p < 0.05$ unpaired two-tailed Student's *T*-test. Data represented as mean ± SEM from $n = 4$ pituitaries or 5 pituitaries for *p16^INK4a* per genotype). AL anterior lobe, IL intermediate lobe, MZ marginal zone, PL posterior lobe. Scale bars in **h**, **n** & **p** represents 200 μm.

We also observed an increase in apoptotic cells in the *CAG:Cre;Braf^Q241R/+* pituitaries compared to Wt (Supplementary Fig. 25). Hence, expression of Braf p.V600E results in an increased apoptosis of the Sox2+ve progenitor stem cell pool, and when coupled with a significant decrease in proliferation, a severe hypoplasia of the anterior pituitary occurs in the *Prop1:Cre;Braf^V600E/+* mutants.

## Discussion
In this manuscript, we report the association of SOD and CFC syndrome in patients harbouring activating mutations in *BRAF*. Hormone deficiencies, such as GH deficiency and delayed puberty, have been reported in patients with CFC, along with some endocrine abnormalities[1,19]. However, the pathogenesis underlying the hormone deficiencies with the link between

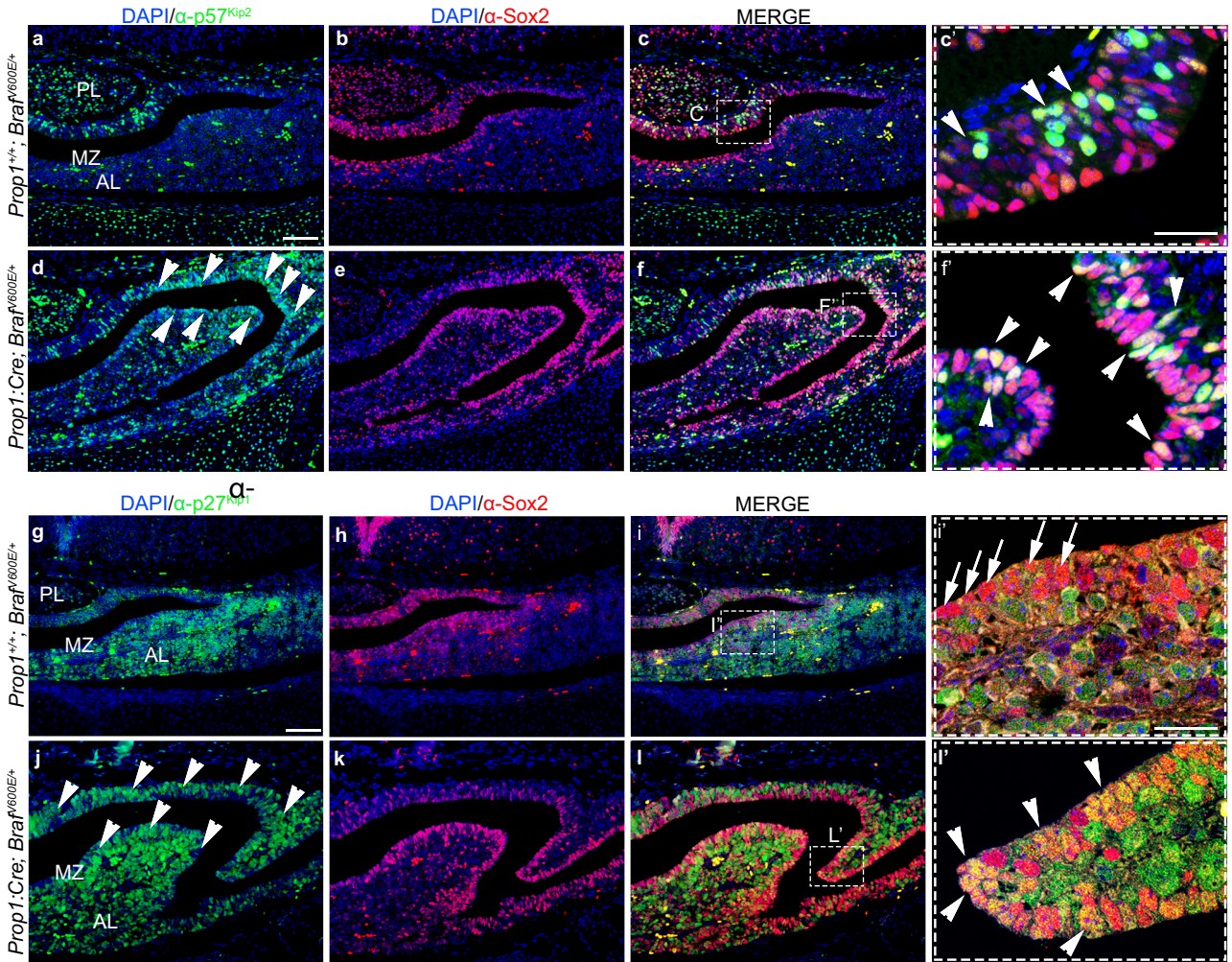

**Fig. 9 Activation of ERK/MAPK pathway by expression of *Braf^{V600E}* results in increased expression of the cell cycle inhibitors p57^{Kip2} and of p27^{Kip1} in the Sox2+ve stem cells at E18.5. a–l** Coronal sections through the pituitary gland at E18.5 of Wt (**a–c, g–i**) and *Prop1:Cre;Braf^{V600E/+}* (**d–f, j–l**). Double immunofluorescence against cell cycle inhibitor p57^{Kip2} (green, **a–f**) and p27^{Kip1} (green, **g–l**) with the pituitary stem cell marker Sox2 (red, **a–l**). The cell cycle inhibitor p57^{Kip2} was found to be upregulated in the *Prop1:Cre;Braf^{V600E/+}* pituitaries (arrowheads in **d**) compared to the Wt (**a**). **c′, f′** Merged enlarged images of squared areas in **c** and **f** reveal increased p57^{Kip2} immunoreactivity co-localising with Sox2 (arrowheads in **f′**) in the *Prop1:Cre;Braf^{V600E}* mutant pituitaries compared to Wt (arrowheads in **c′**). Expression of p27^{Kip1} (arrowheads in **j**) is observed in the marginal zone (MZ) of the mutant pituitaries compared to Wt (**g**). Confocal merged images of the marginal zone revealed co-localisation of Sox2 with p27^{Kip1} in the mutant *Prop1:Cre;Braf^{V600E/+}* pituitaries (yellow nuclei, arrowheads in **l′**), while no co-localisation of p27^{Kip1} and Sox2 was seen in Wt pituitaries (arrows in **i′**). **i′, l′** are enlarged images of the squared areas in **i** and **l** respectively. Images are representative of three embryos per genotype. AL anterior lobe, MZ marginal zone, PL posterior lobe. Scale bars in **a** and **g** represent 200 μm. Scale bars in **c′** and **i′** represent 25 μm.

RASopathies and developmental abnormalities of the HP axis leading to CH have not been previously established. In this study, we characterise the functional consequences of one genetic variant in BRAF, BRAF p.F468S, which has been previously reported but not functionally characterised. We also report more detailed functional analyses of the less well characterised BRAF p.T241P and BRAF p.G469E mutants, which occur in both Leopard syndrome and CFC[25,27,28]. Phosphoproteomic analyses of these genetic variants demonstrate that all the genetic variants are indeed pathogenic, with the BRAF p.T241P, BRAF p.F468S and BRAF p.Q257R variants resulting in similar phosphopeptide enrichment and clear over-activation of the ERK/MAPK pathway. However, the BRAF p.G469E genetic variant showed relatively modest activation of the ERK/MAPK pathway in our phosphoproteomic analyses, indicating that this is a mild activator of the pathway, coinciding with a milder clinical phenotype with no cardiovascular or HP phenotypes, although the latter

could still evolve given that the growth pattern of the proband is abnormal.

Short stature may be multifactorial in CFC patients and other RASopathies, for example, due to poor feeding, as well as gastrointestinal and cardiac defects, which may mean that endocrine evaluation is often not undertaken in CFC patients. Our murine transgenic experiments show that the MAPK pathway is essential for pituitary gland development, with activating mutations leading to CH and therefore patients with CFC should be screened for pituitary hormone deficiencies. We show that activation of the ERK/MAPK pathway by expressing Braf p.V600E only in the pituitary gland (Prop1+ve pituitary progenitors cells) or the knock-in allele of the most common human CFC-causing mutation, the hBRAF p.Q257R (*CAG:Cre;Braf^{Q241R/+}*), results in clear hypopituitarism with a decrease in the cell lineage determination factors Pit1 and Sf1, required for terminal differentiation of somatotrophs (GH+ve), thyrotrophs (TSH+ve), lactotrophs (PRL+ve) and

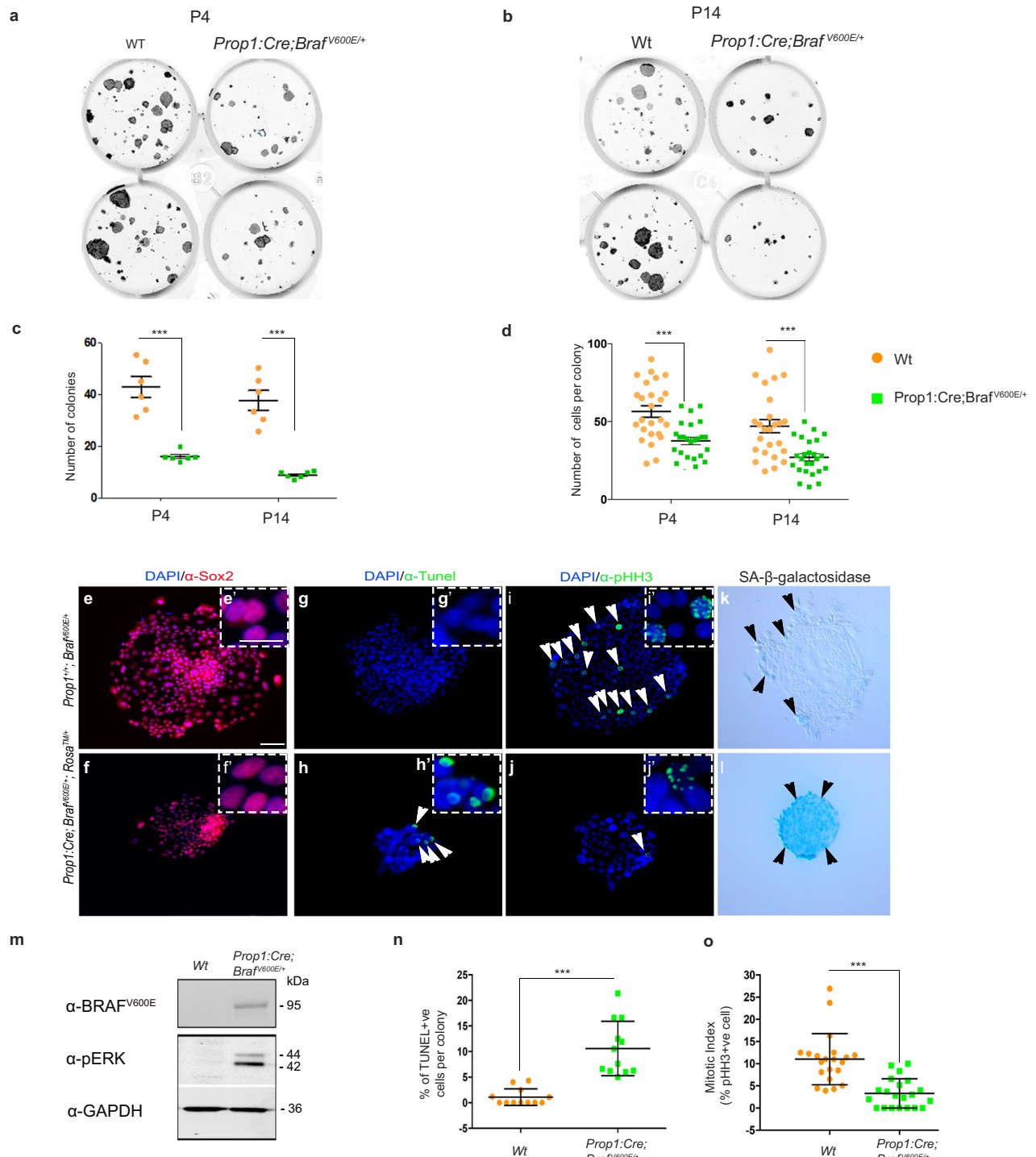

**Fig. 10 Expression of Braf p.V600E in postnatal pituitary stem cells leads to decreased proliferation and increased apoptosis in vitro.** PSC cultures from Wt and *Prop1:Cre;Braf^{V600E/+}* at postnatal stage P4 (**a**) and P14 (**b**) reveal a significant decreased capacity in colony formation (**c**) and number of cells per colony (**d**) in mutant PSCs compared to Wt. The ability of the mutant PSCs to form colonies diminishes over time from P4 to P14 (**c**). Immunostaining with the PSCs marker Sox2 revealed that all the cells in culture are Sox2+ve (**e**, **f**). TUNEL immunofluorescence revealed a significant increase in apoptotic cells in the mutant PSC colonies (arrowheads in **h** and quantification **n**) while almost no apoptotic cells were seen in the Wt colonies (**g**, **n**). Immunofluorescence against pHH3 revealed a substantial decrease in pHH3+ve cells in the mutant PSCs (arrowhead in **j**) compared to Wt (arrowheads in **i**). **o** Quantification of the number of pHH3+ve cells per colony shows a significant decrease in the mitotic index in the mutant PSC colonies compared to Wt. The *Prop1:Cre;Braf^{V600E/+}* mutant colonies express the senescence SA-β-galactosidase (arrowheads **l**) while only a few positive cells were detected in Wt (arrowheads in **k**). Western blotting of PSC lysate revealed expression of Braf p.V600E resulting in increased pERK in the *Prop1:Cre;Braf^{V600E/+}* mutant PSCs compared to Wt. *** statistically significant $p < 0.001$, unpaired two-tailed Student's *T*-test, data represented as mean ± SEM (number of colonies and cells per colony of three mutants and three Wt from three independent experiments performed in triplicates **c**, **d**); **n** number of TUNEL+ve cells per colony of 12 colonies from three mutants and three Wt; **o** number of pHH3+ve cells of 21 colonies from three mutants and three Wt. Images are representative of three independent experiments. Scale bar in **e**′ and **e** represent 50 and 10 μm respectively.

gonadotrophs (LH+ve and FSH+ve). Importantly, these phenotypes partially recapitulate endocrinopathies reported in our CFC probands in this study and in association with other reported RASopathies[1,19,53–55].

Interestingly in both our murine models, $Prop1:Cre;Braf^{V600E/+}$ and $CAG:Cre;Braf^{Q241R/+}$, activation of ERK/MAPK signalling results in an increase in PRL+ve and ACTH+ve cells during development. The increase in lactotrophs upon activation of the MAPK pathway in our mutants is consistent with several in vitro and in vivo studies[56–60]. Indeed, persistent activation of the ERK/MAPK pathway in rat GH4 pituitary somatotrophs and lactotrophs, by either addition of exogenous epidermal growth factor (EGF) or expression of oncogenic RasV12, results in increased secretion of PRL. Additionally, EGF treatment of postnatal pituitaries has been shown to drastically increase the proportion of lactotrophs and PRL secretion through increased ERK/MAPK signalling[56,60]. Of interest, a CFC patient carrying a heterozygous mutation in $MEK1$ has been reported to have GH deficiency with hyperprolactinemia[61].

Our results demonstrate that activation of the ERK/MAPK pathway results in a significant increase of TPit+ve and corticotroph cells. This is in line with multiple reports in which activation of the MAPK pathway has been shown to be required for the transcriptional activation of the $Pomc$ gene in ACTH+ve corticotrophs[62,63]. Mutations in hUSP8, which lead to increased EGFR signalling via activation of the MAPK pathway, result in increased $Pomc$ expression in corticotroph adenomas[64], and Fukouka et al.[65] have shown that $Pomc$ promoter activation is dependent on MAPK and can be inhibited by the EGFR blocker Gefitinib.

Fibroblast growth factors Fgf8, Fgf10 and Fgf18 signal through the MAPK pathway, and their expression in the pituitary organiser, the infundibulum, is essential for pituitary progenitor proliferation and anterior pituitary formation[66,67]. Our data partially agree with a recent study in which the ERK/MAPK pathway was activated using two different alleles, namely $Kras^{G12D}$ and $Braf^{V600E}$, under $Hesx1$ regulatory elements[68]. In this report, activation of the MAPK pathway also resulted in increased ACTH+ve cells, a decrease in the cell lineage determination factors Pit1 and Sf1, and an increase in the number of TPit+ve cells, which are all consistent with our results. Activation of either $Kras^{G12D}$ or $Braf^{V600E}$ in Hesx1+ve cells resulted in perinatal lethality, with none of the mutant pups surviving to birth[68]. Hence, the early lethality seen in these mutants precluded the study of the effect of Braf p.V600E expression in pituitary progenitors postnatally. In our experiments, embryos that expressed Cre and activated Braf p.V600E in the CNS, developed severe brain abnormalities and perinatal death. Hence we designed our experiment to select only pituitary-specific activation of the $Braf^{V600E}$ allele. We found that pups carrying the $Braf^{V600E}$ allele only in the pituitary gland survive birth and do not develop pituitary tumours but instead develop hypopituitarism.

Previous studies have identified somatic BRAF p.V600E mutations as drivers of two pathologically distinct pituitary tumours, namely the non-secreting benign pituitary tumour known as papillary craniopharyngioma (PCP)[16,17], and more recently ACTH-secreting pituitary adenomas leading to Cushing's disease[18]. Hence, BRAF p.V600E can lead to two pathologically distinct types of pituitary tumours, most probably depending on the pituitary cell type of origin from which the mutation arises. Our murine data suggest that expression of Braf p.V600E in embryonic pituitary progenitors/stem cells does not lead to pituitary tumours. One possibility is that the expression of Braf p.V600E alone is not sufficient to cause tumours. This is in agreement with several studies that show that activation by Braf p.V600E alone promotes cell growth inhibition,

lack of terminal differentiation and apoptosis[46,47,49,51]. Furthermore, PCPs affect mainly adults, and consist of undifferentiated cells, which is at variance with the highly differentiated corticotroph and lactotroph populations of our $Prop1:Cre;Braf^{V600E/+}$ mutants. Therefore, it is plausible that PCPs require a second mutational hit in either another oncogene or a tumour suppressor. Alternatively, in order for BRAF p.V600E to lead to tumour formation, the mutation may need to occur in a differentiated pituitary cell or adult pituitary stem cell rather than in an embryonic pituitary progenitor.

The early postnatal death of our $Prop1:Cre;Braf^{V600E/+}$ mutant mice at around weaning may be attributed to severe hypopituitarism with complete lack of TSH. Thyroxine deficiency in mice has been linked to postnatal lethality in several studies, and thyroxine has been shown to be essential for survival after 6 weeks of age and post-weaning to independent life[69,70]. The CFC-causing mutation ($CAG:Cre$; $Braf^{Q241R/+}$) in a C57BL/6 genetic background shows terminal differentiation deficiencies with a decrease in GH, TSH, LH and FSH, which is in accordance with our CFC patients. Due to cardiovascular abnormalities associated with CFC syndrome, these mice die perinatally, however, in a CD1 genetic background these animals survive the cardiovascular abnormalities and develop dwarfism with low IGF1 levels, further reinforcing the effect of the genetic background on the CFC phenotype[71,72].

Our study has specifically investigated the role of Braf in the murine pituitary, where activation of the MAPK pathway through expression of $Braf^{V600E}$ or $Braf^{Q241R}$ alleles leads to pituitary hypoplasia and CH. Expression of $Braf^{V600E/+}$ in pituitary progenitors leads to a transient increased proliferation of the Sox2 +ve PSCs, leading to enlargement of the marginal zone. However, the proliferative capacity of the mutant pituitaries significantly decreases later in development, with pituitaries becoming hypoplastic and favouring differentiation into ACTH+ve and PRL+ve cells. The Sox2+ve pituitary stem cells aberrantly express ACTH and PRL and fail to normally differentiate into GH-, TSH-, FSH- and LH-producing cells. Moreover, activation of the ERK/MAPK pathway leads to increased expression of the senescence-associated markers $p16^{INK4a}$, p21, SA-β-Galactosidase, and increased expression of the cell cycle-dependent kinase inhibitors $p57^{Kip2}$ and $p27^{Kip1}$, leading to cell growth arrest and increased apoptosis of the Sox2+ve progenitor/stem cell pool. Apoptosis of the Sox2+ve pituitary stem cells coupled with cell growth arrest leads to depletion of the stem cell pool and pituitary hypoplasia, rather than tumour formation.

The patients reported in this study harbour activating mutations, and patients 1–4 clearly exhibit varying degrees of hypopituitarism. The rather poor response to GH treatment observed in our patients in the face of GH deficiency suggests that there may be a co-existing GH insensitivity, or altered function of growth plate chondrocytes, as has been previously described in RASopathies. Patient 1 showed evidence of hypogonadotropic hypogonadism. Of note, IGF1 was low in patients 3 and 5 in spite of normal GH concentrations in response to provocation, and this may reflect neurosecretory dysfunction of GH secretion, as has also previously been documented in RASopathy patients. Three of our patients (patients 2, 3 and 4) manifested exuberant LH and FSH responses to GnRH stimulation, with patients 2 and 3 needing sex steroids to progress through puberty. Our studies do not exclude a hypothalamic contribution to the phenotype in humans, as $Braf/BRAF$ is expressed in both the hypothalamus and the pituitary; however, our study has focused on demonstrating the role of BRAF within the pituitary gland.

To date, there has been no molecular explanation underlying the association between childhood onset hypopituitary disorders

such as SOD and BRAF variants. Unifying features include GH deficiency, with the evolution of other pituitary abnormalities such as TSH deficiency, which parallels the phenotype of both murine models (*Prop1:Cre;Braf*$^{V600E}$) and the human CFC-causing mutation (*CAG:Cre;Braf*$^{Q241R/+}$). Individuals with CFC syndrome should therefore be screened for pituitary abnormalities and hypopituitarism, as these are associated with further morbidity and, if undiagnosed, potential mortality. Our findings show a direct and vital role for BRAF in the development of the HP axis in both mice and humans, and implicate for the first time *BRAF* mutations found in RASopathies as an underlying cause of congenital endocrine deficiencies in humans, thereby explaining previously described endocrinopathies in CFC/RASopathies. Hence, patients with RASopathies should be closely monitored for endocrine deficiencies early in life. Our work also reveals that BRAF and components of the MAPK pathway are potential novel candidate genes for congenital pituitary disease, such as SOD, or isolated or CPHDs, and thus mutations in components of the MAPK pathway could be mutated in CPHD. In conclusion, our murine models illustrate a role for BRAF and, more generally, the MAPK signalling pathway in pituitary development, and explain the underlying mechanism by which activating mutations in components of the MAPK pathway can lead to hypopituitarism.

## Methods

**Animals.** All experiments were conducted under the regulations, licenses and local ethical review of the UK Home Office Animals (Scientific Procedures) Act 1986 and are described and QM-AWERB Ethical committee. The transgenic lines *Rosa26*$^{CAGLoxpSTOPLoxpTdTomato}$ (stock #007905), *Braf*$^{V600E/+}$ (stock #017837) were obtained from the JAX lab and have been previously described[31,33]. The *Prop1:Cre* transgenic line[32] was kindly provided by Shannon Davis and Sally Camper. The *CAG:Cre;Braf*$^{Q241R/+}$ mice were provided by Shin-ichi Inoue et al.[34]. Animals were kept in 12 h light/12 h dark cycle, with constant supply of food and water, temperatures of 65–75 °F (~18–23°C) with 40–60% humidity.

**Patient recruitment.** Patients with CFC were recruited to the study, and Sanger sequencing was performed in regional accredited Genetics laboratories. Ethical committee approval was obtained from the UCL Great Ormond Street Hospital for Children Joint Research Ethics Committee (09/H0706/66). Informed written consent was obtained from all patients and/or parents. The human embryonic and foetal material was provided by the Joint Medical Research Council (MRC)/Wellcome Trust HDBR Resource (www.hdbr.org) with approved Research Ethics Committee 18/NE/0290 and 18/LO/0822.

**Whole-exome sequencing and alignment.** Whole-exome capture and sequencing was performed at BGI (Shenzhen, China) using SureSelect Human All Exon v6 60 Mb kit (Agilent Technologies, Santa Clara, CA, USA) and BGISEQ-500 platform (Illumina, San Diego, CA, USA). Sequencing reads were aligned with Burrows-Wheeler Aligner (BWA) v0.7.17[74] to human genome build 38 (GRCh38.p1) not including alternate assemblies (GCA_000001405.15_GRCh38_no_alt_analysis_set.fna). Read duplicates were marked with Sambamba[75].

**Variant calling and annotation.** Variant calling across exome capture target regions with 100 bp padding was performed using Genome Analysis Toolkit (GATK) v4.0.3.0[76,77] according to the best practices workflow for joint (multi-sample) calling[78]. The resultant variants were normalised and decomposed using Bcftools v1.8 (https://github.com/samtools/bcftools) and annotated with ANNO-VAR[79]. All variants in the genes previously associated with hypopituitarism, SOD, and CFC were assessed for pathogenicity. In order to exclude any other reported pathogenic variation in the exome we also examined all variants listed as 'pathogenic' and 'likely pathogenic' in the ClinVar database (v.2018-10-28) and variants annotated as 'pathogenic' and 'likely pathogenic' by InterVar.

**Plasmids and site-direct mutagenesis.** The full-length cDNA *hBRAF* (NM_004333.4) clone in MAM pCR4-TOPO vector was provided by www.hdbr.org. *Hin*dIII and *Not*I restriction sites were introduced by PCR and products subcloned in the pcDNA3.1 (+) (Addgene). Mutagenesis was performed using QuikChange II XL Site-Directed Mutagenesis Kit (Agilent Technologies) according to the manufacturer's instructions. Mutagenesis primers are indicated in the Supplementary Table 4. All mutations were confirmed by Sanger sequencing.

**Cell culture and western blotting.** HEK293T cells were grown in Dulbecco modified Eagle' medium (DMEM) supplemented with 10% FBS. Cells were seeded

in 24-well plates at $1.75 \times 10^5$ cells/well 24 h before transfection. Cells were transfected with equal amounts (200 ng) of Wt or mutant p.T241P, p.Q257R, p. F468S, p.G469E and pV600E hBRAF plasmids using Lipofectamine 2000 (Life-Technologies) according to the manufacturer's instructions. Cells were harvested 24 h after transfection in a lysis buffer [50 mM Tris-Base (pH 7.6), 150 mM NaCl, 1% Triton X-100] implemented with protease inhibitors (Complete Mini, EDTA-free tablets, Roche) at 1:6 ratio with the total volume and 1% phosphatase inhibitor Cocktail3 (Sigma-Aldrich)]. Bradford assay was used to quantify protein (Pierce BCA Protein Assay Kit, Thermo Scientific). Western blot membranes were incubated overnight at 4 °C with primary antibodies (Supplementary Table 6). Membranes were analysed using Odyssey 2.1 Imaging System (LI-COR Biosciences). Experiments were independently repeated nine times and the statistical analysis was performed using one-way ANOVA.

**Phosphoproteomics.** Cells were washed twice with PBS supplemented with 1 mM Na$_3$VO$_4$ and 1 mM NaF, lysed in urea buffer (8 M urea in 20 mM in HEPES pH 8.0 supplemented with 1 mM Na$_3$VO$_4$, 1 mM NaF, 1 mM Na$_4$P$_2$O$_7$ and 1 mM sodium β-glycerophosphate) and stored at −80 °C. Cell lysates were further homogenised by sonication, insoluble material was removed by centrifugation and protein in cell extracts was quantified. Following described procedures, 250 µg of protein was reduced, alkylated and digested with trypsin. Peptide solutions were desalted with Oasis cartridges and phosphopeptides enriched using TiO$_2$ as previously reported[80]. Phosphopeptide pellets were re-suspended in reconstitution buffer (20 fmol/µl enolase in 3% ACN, 0.1% TFA) and loaded onto an Orbitrap Q-Exactive Plus mass spectrometer (Thermo Fisher Scientific) operated with a parameter setting previously described[80]. Peptide identification from MS data was automated with Mascot Daemon 2.5.0. Searches were performed against the SwissProt Database (uniprot_sprot_2014_08.fasta) using the parameters described in [81]. Pescal (v01)software was used for label-free peptide quantification[80], and undetectable peptides were assigned a value equal to the lowest detected intensity across sample divided by 10. Values of two technical replicates per sample were averaged and intensity values for each peptide were normalised to total sample intensity. Differences in peptide phosphorylation between Wt and BRAF variants were reported as fold over Wt and statistical significance for those changes was assessed using unpaired two-tailed *t*-test. Kinase activities from phosphoproteomics data were inferred by KSEA as described before[80].

**Immunohistochemistry, immunofluorescence, and in situ hybridisation.** Immunostaining was performed by deparaffinisation of the sections followed by rehydration through decreasing ethanol dilutions. Heat-induced antigen retrieval was performed with a microwave in 10 mM sodium citrate buffer (pH 6). Samples were left to cool and incubated for 1 h in blocking buffer [1 PBS, 0.1% Triton X-100, 5% Normal Goat Serum (Vector Laboratories)]. Primary antibodies and their concentration are listed in Supplementary Table 6. Staining was achieved using DAB Peroxidase Substrate Kit (Vector Laboratories; SK-4100). The colorimetric reaction was stopped with water and the sections were counterstained using haematoxylin (Sigma-Aldrich). For immunofluorescence, conjugated secondary antibodies Alexa Fluor 568 or 488 were used, or a biotinylated secondary followed by streptavidin. Sections were mounted with Vectashiled containing DAPI (Vector Laboratories). Images were acquired with Leica or confocal LSM 880 laser scanning confocal microscope with AiryScan. Figures were generated with Adobe Photoshop CS6. The MI is the percentage of pHH3-positive cells compared to total number of cells (average counts from three different sections, separated approximately by 100 µm, per each embryo/pituitary with a minimum of $n = 5$–8 per genotype and stage). Caspase represent number of positive cells per section with an average of three sections per pituitary/embryo. In situ hybridisation was performed by adapting the protocol from [53] and described before in[73,82]. In short, slides were deparaffinised, rehydrated and fixed with 4% PFA. Slides were incubated with proteinase K, followed by a second fixation with 4% PFA and finally incubated with 0.1 M triethanolamine and 0.1% acetic anhydride (Sigma). Hybridisation was achieved by an overnight incubation with 100 ng of the digoxigenin-labelled probe at 65 °C. Sections were washed in 0.1 M Tris-HCl Buffer (pH = 7.5) followed by an overnight incubation at 4 °C with anti-Dig antibody (Sigma-Aldrich). Signal detection was achieved by colorimetric reaction using 4-nitro blue tetrazolium chloride solution (NBT; Sigma-Aldrich) and 5-bromo-4-chloro-3-indolyl phosphate disodium salt (BCIP; Sigma-Aldrich). The digoxigenin-labelled antisense probes *hBRAF, Pomc1, Pit1, Prop1, Lhx3, Pitx1, Pax7, Sf1, p16*$^{INK4a}$*, p27*$^{Kip1}$*, p57*$^{Kip2}$ and *p21* were generated from plasmids containing either a portion or full-length cDNA of each gene obtained from Source Bioscience, H.D.B.R., A. McMahon; M. Rosenfeld; Sally Camper; Leonardo Guasti, Andreas Kispert and Peter Gruss[83,84], respectively. In utero BrdU (5-bromo-2′-deoxyuridine; Sigma) was performed by IP injection of pregnant females at a final concentration of 100 mg/kg; 2 h after injection embryos were dissected and fixed. At least three embryos per genotype were used for each gestational stage. Cell counts were performed using ImageJ software, and graphs and statistics using Graph-Pad Prism v.9.

**RT-qPCR gene panel and primer design.** RNA expression levels of *Cdkn2a* (p16$^{INK4a}$), *Cdkn1a* (p21), *Cdkn1b* (p27$^{Kip1}$), *Cdkn1c* (p57$^{Kip2}$) genes were

analysed by RT-qPCR. RT-qPCR was performed using QuantiTect SYBR Green PCR Kit (Qiagen) according to the manufacturer's protocol and analysed with Stratagene (Agilent Technologies). A comparative Ct method ($2^{-\Delta\Delta CT2}$ method) was used to compare the mRNA expression levels of genes of interest normalised to GAPDH. Differences in mRNA expression levels were compared by using Student' T-tests. Primers are shown in Supplementary Table 5.

**Pituitary stem cell culture and SA-β-galactosidase**. PSCs were cultured from murine AL incubated for 2 h in enzyme mix [0.5% w/v Collagenase, 50 µg/ml DNase, 2.5 µg/ml Fungizone, trypsin 0.1% in Hank's Balanced Salt Solution] and mechanically dissociated into single cells. In all, 10,000 cells/well were plated in 12-well plates and cultured in stem cell-promoting media [Ultraculture Medium (Lonza), supplemented with 5% FCS (Sigma), 1% penicillin/streptomycin (P/S: Fisher), 1% glutamax (Fisher), 20 ng/ml basic fibroblast growth factor (R&D) and 50 ng/ml cholera toxin (Sigma)]. Media was changed every 48 h and cultures were maintained for 8 days. SA-β-Gal staining: staining on PSC cultures was performed according to the manufacturer's instructions (Cell Signalling kit (#9860)).

**Statistics and reproducibility**. Statistical analyses were performed using Prism 6 and 9 software (GraphPad). The number of independent experiments and of replicates ($n$) is indicated in each the figure legends. Unless stated otherwise, at least three biological independent replicates were performed for each panel and came from at least three independent experiments. When appropriate, normalisation of the data was performed within each independent experiment.

**Reporting summary**. Further information on research design is available in the Nature Research Reporting Summary linked to this article.

## Data availability

The authors declare that all the data supporting the findings herein are included in the article (or Supplementary materials) and available from the corresponding author (C.G. M.) upon reasonable request. All mass spectrometry proteomics data have been deposited to the ProteomeXchange Consortium via the PRIDE (https://www.ebi.ac.uk/pride/archive/projects/PXD018190) partner repository with the dataset identifier PXD018190. The source data underlying graphs and un-cropped gels in the manuscript main figures and supplementary materials are provided as a Source Data file. The exome sequencing data that support the findings are not publicly available due to information that could compromise the research participant's privacy/consent. A reporting summary for this article is available as a Supplementary Information file.

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

## Acknowledgements

We apologise to the authors whose work we were unable to reference due to space constraints. We thank Simon Rhodes and J. Drouin for providing the PIT1 and TPIT antibodies. The human embryonic and foetal material was provided by the Joint MRC/Wellcome Trust (grant# MR/R006237/1) Human Developmental Biology Resource (www.hdbr.org). The research is funded by Action Medical Research (GN 2272) and BTLC (GN 417/2238 and MGU0551). R.T. and J.B. were funded by MRC-CRTF (MR/P018459/1 and MR/S037896). V.S., M.L.V., A.G. and C.G.-M. were funded by Early Career Fellowship from the Medical College of Saint Bartholomew's Hospital Trust. P.R.C. and P.C. were supported by grants from CRUK (C15966/A24375), BTLC (297/2249) and BBSRC (BB/M006174/1). R.E.J.B. was funded by NIHR from an academic Clinical Lectureship award. Y.A. was supported by Japan Agency for Medical Research and Development under Grant (JP18ek0109241 and JP20ek0109470), and S.I. by JSPS KAKENHI Grant Number 18K07811. M.T.D. was funded by the Great Ormond Street Hospital (GOSH) Children's Charity, the NIHR GOSH BRC, and partially funded by the Medical Research Foundation (MRF-099-0002-RG-UCLIC). The analysis performed by GOSgene in this study is in part supported by the National Institute for Health Research (NIHR), GOSH and the Biomedical Research Centre (BRC). The views expressed are those of the author(s) and not necessarily those of the NHS, the NIHR or the Department of Health.

## Author contributions

C.G.M. and M.T.D. conceived the work and obtained funding. C.G.M., L.C.G., N.K., M.L.V., J.G.N., A.G., P.C., V.S., R.T., E.M., J.B., and F.A.-J. performed experiments and analysed data. R.E.J.B. collected and summarised patient, clinical and laboratory data. S.W.D. and S.A.C. provided the *Prop1:Cre*; and S.-i.I. and Y.A. provided the *CAG:Cre; Braf^Q241R/+* transgenic line. J.H.D. and I.K.T. first identified the association between SOD and CFC in two patients. J.H.D., I.K.T. and W.H. provided patient and clinical data. E.F.G. supervised N.K. and analysed data. ICAFR provided funds for M.T.D. and L.C.G. M.L.V., P.C. and P.R.C. performed phosphoproteomic experiments. C.G.M. wrote the paper with input from all the authors.

## Competing interests

The authors declare no competing interests.
