## [Peer Review File · Nature Communications]

In the material and methods, pituitary stem cell cultures are said to be established from Sox2CreERT2/+; BrafV600E/+ mutants. We have addressed this point by using double immunofluorescence against TUNEL (marker of apoptotic cells) and Sox2. We have extended our apoptosis analysis to postnatal day (P) 5 and we have shown that at both E18.5 and P5 there is an increase in TUNEL positive cells that co-express Sox2 (Supplementary Figure 24). We have also performed immunostaining of pituitaries from Sox2+ cells cultured from control P4 and P44 mutant and wild type pituitaries at E18.5 and P5 on the pituitary Sox2+ve PSCs. This suggests increased apoptosis rate selection of precursors of the primary dysmorphism and that the dysmorphism does not drive the cell fate selection process. In previous studies (using a active CASPASE antibody: Supplementary Figure 23 & 25) and show that the Sox2+ve stem cells of Braf mutant pituitaries undergo increased apoptosis both *in vivo* and *in vitro*. In line with this, we observed increased TUNEL+ve cells from E18.5 to P5, further demonstrating increased apoptosis of the stem cell pool in the Braf mutants. We have examined the expression pattern of the early developmental transcription factors *Prop1*, *Lhx3* and *Pitx1* by *in situ* hybridisation at E11.5 and at E13.5 in the *Prop1*Cre;Brf^{V600E/+} and the *CAG*Cre;Brf^{Q241R/+} models (Supplementary Figure 24). We did not find discernible differences in *Prop1* expression (Supplementary Figure 24). Therefore, it is not very clear that an increase in progenitor proliferation does alter the expression of the major factor explaining the hypoplasia, as suggested by the authors. The function of BRAF as oncogene is well known and mutations have been reported in various types of cancer. BRAF is part of the RAS/MAPK signaling pathway which is associated with congenital syndromes termed RASopathies. The manuscript by Gualtieri et al. now provides similar number of forming colonies between mutant and Wt. We think this may reflect subtle differences in the pituitary development. Based on different mutations identified in 3 patients with Septo-optic dysplasia associated with CFC syndrome, they show that these mutations lead to increased kinase activity and activation of the ERK/MAPK pathway, which is followed by a downregulation of cell cycle inhibitors p57^{Kip2}, the senescence markers p21^{INK4}, p16^{INK4} and the SA- β -galactosidase. However, in these hypoplasia, the onset is evident at Q25 postnatal stages, a knock-out of the BRAF pathway also in Figure 10. The proliferation of the pituitary stem cells as well as the proliferation capacity of BrfV600E expressing PSCs *in vitro*, we have expected to be similar to control. We have performed assays to postnatal stages to measure the number of colonies formed in the pituitary stem cell culture. We found that the number of colonies and the number of cells per colony are significantly reduced compared to Wt, indicating that the proliferative capacity of mutant PSC is compromised (Supplementary Figure 22 A-D). Importantly, immunostaining with the apoptotic marker TUNEL identified a significant increase in apoptosis in the mutant BrfV600E PSCs compared to Wt (Supplementary Figure 22 C-D). We also observed a reduction in mitotic index of the BrfV600E mutant PSCs compared to Wt and the mutant mitotic index described in the text, it is not entirely clear if this index is increased, decreased or unaltered. Add more positive staining of the senescence marker SA- β -galactosidase.

3. Some spelling errors: whether (407), distinct (415), die in the (432), apoptosis (444). It might be helpful to add statistical significance to the growth chart in Fig 3. We have the MAPK pathway activity expression of BrfV600E and the statistical significance of proliferation and increased apoptosis of the Sox2+ve PSCs. Our new data strengthen our results *in vivo* and *in vitro* results (Figure 8, 9, 10 & Supplementary Figure 23, 24, 25) in which we report that expression of BrfV600E leads to increased cell cycle inhibitors, increased mitotic index, and increased apoptosis leading to severe hypoplasia of the pituitary gland. **Line 345 needs the word induction or upregulation after 4.6 fold.** Thank you, upregulation has now been added.

In addition, it would have been useful to examine patterns of MAPK activation to better characterise the consequences of the expression of the Braf mutants

We have used immunofluorescence using an antibody against the MAPK effector α -PERK as a read out of MAPK pathway activity. Expression analysis in Wt and *Prop1*Cre;Brf^{V600E/+} mutant pituitaries revealed expression of pERK during early pituitary development both in Wt and mutant, indicating that at these early stages RP cells are responsive to MAPK activation. In the mutant pituitaries, pERK positive cells remain at E13.5 and E15.5 and are localised in the marginal zone where the progenitors are located. At E18.5, many pERK positive foci are seen along the lining of the marginal zone (Supplementary Figure 26).

Indeed, the seemingly unperturbed emergence of corticotrophs in mutants may be explained by the activity of Prop1-Cre which first becomes detectable at 11.5dpc, as the first corticotroph differentiate so maybe too late to perturb their emergence.

This is a very interesting point which we did not address in our first submission. In order to identify if the Cre activity from the *Prop1:Cre* transgenic line occurs before or after the appearance of the Pomc cells, leading to unperturbed emergence of corticotrophs, we have taken two experimental approaches. **Firstly:** we have studied the expression of Tomato from the *Prop1:Cre;Braf^{V600E/+};RosaTM* during early Rathke's Pouch development. At E10.5, we identify a "salt and pepper" expression pattern, with few Tomato positive cells, indicating activity of Cre at this stage (Supplementary Figure 9, A-D). Immunostaining with Tomato reveals that by E12.5, all pituitary cells appear Tomato positive (Supplementary Figure 9, J-K, K'). We then looked at Pomc expression by IHC and found Pomc+ve cells within the ventral diencephalon but not in the RP at E11 (arrows in E & G Supplementary Figure 9). We detect Pomc+ve cells in the anterior lobe by E12.5 (arrowhead in H, Supplementary Figure 9); by this stage the whole of the pituitary is labelled with Tomato (Supplementary Figure 9, J-K, K').

Secondly: To further assess if the Cre activity marks the emerging Pomc+ve cells, and the other pituitary cell committed lineages (TPit, Pit1, Sf1), we have used the *Rosa^{CAGLxpSTOPLxpTomato}* allele from the *Prop1:Cre;Braf^{V600E/+};RosaTM* as a cell lineage tracer. Cells that express Cre will recombine the LxpSTOPLxpTomato leading to expression of the Tomato protein in the cells where Cre removes the stop silencing cassette and all its descendants. Double immunostaining staining with -Tomato and Pomc identified that all the Pomc cells co-express Tomato, indicating that Cre activity marks newly emerging Pomc cells at E12.5 and E15.5 (arrowheads in K' & N', Supplementary Figure 9). We extended our study to other cell lineages at E15.5 by double immunostaining for Tomato and TPit, Pit1 or Sf1. We identified co-localisation of the 3 cell lineage commitment markers with Tomato (arrowheads in Supplementary Figure 10). Together, our results demonstrate that Cre-activity from the *Prop1:Cre* transgenic line starts early in Rathke's Pouch development at E10.5 and labels all of the pituitary gland by E12.5, marking all the lineages including the early emerging corticotrophs. These results are in line with the cell lineage tracing experiments done using the *LxpSTOPLxpLacZ* reporter crossed with *Prop1:Cre* transgenic line, showing that all *Prop1:Cre* descendants give rise to all anterior pituitary terminally differentiated cells.

Davis, S.W., S.A. All Hormone-Producing Cell Types of the Pituitary Intermediate and Anterior Lobes Derive From Prop1-Expressing Progenitors. *Endocrinology* **157**, 1385-1396 (2016).

"Could the switch in proliferative activity correlates with a decrease in MAPK activity?"

Based on our immunostaining with the activated MAPK readout (α -pERK), the pituitary progenitors in Wt express pERK at early stages of pituitary development and this staining decreases over time (Supplementary Figure 26). Hence, as the reviewer suggests, during normal development early progenitors "are more responsive" or "competent" to the activated MAPK pathway, which may explain the initial hyperproliferation phenotype seen in our mutants. However, pERK positive immunostaining persisting beyond the "normal developmental stages" in the marginal zone at E15.5 and E18.5 in *Braf.V600E* mutants (Supplementary Figure 26), indicating that there is no reduction in MAPK activity, but rather MAPK activation continues to later stages. We favour the explanation that sustained MAPK activation after E15.5 leads to cell senescence, increased expression of cell cycle inhibitors by the Sox2+ve PSCs, cell growth arrest followed by a significant increase in apoptosis of the Sox2+ve cells during development.

"Are POMC lineages activating the pathway at the same level than other cell types?"

Our data indicate that increased MAPK signalling favours Sox2+ve differentiation into corticotrophs. Increased TPit expression (Figures 7 & Supplementary Figure 20). We show that once cells commit into Pomc, these cells do not over-proliferate by increased MAPK (BrdU experiment, Supplementary Figures 18 & 19). Hence, the MAPK pathway directly affects the Sox2+ve cells, which fail to differentiate into Pit1 and Sf1 cell lineages. The authors agree with the reviewer that addressing differential levels of MAPK activation by the Pomc lineages compared to other cell types will be very interesting. Equally important would be to identify why Pit1 differentiation is negatively affected by increased levels of MAPK since GH and TSH deficiency are

an important clinical phenotype. However, in order to be able to quantify levels of MAPK activity within the Pomc cells and to compare them to other cell types *in vivo* (in a quantitative fashion), it will require cell sorting of pituitary cell lineages with specific Cre-transgenic lines, such as (Pomc:Cre), (Pit1:Cre) and (Sf1:Cre) crossed to Braf mutants, followed by quantitatively assessing the MAPK downstream effector levels. Although we agree this will be a very interesting study, we think that within the timescale of this revision it will be difficult to achieve these results, as it will require a series of transgenic crosses and proper quantification of MAPK effectors and downstream targets in very few cells.

Detailed comments:

“When mentioned in figures it seems that ACTH should be replaced by POMC, because the intermediate lobe is stained, so the protein detected is the precursor POMC. In many instances in the text Pomc is referred as Pomc1”.

This has now been changed to Pomc in most of the figures and text. Pomc1 has been changed to Pomc. We would like to note that despite the detection of the precursors by the antibody, we do not see an increase of Pax7+ve melanotropes (Supplementary Figure 13) but rather corticotrophs.

“Figure 9: The p27 staining presented here is not convincing”

We have repeated the double immunofluorescence against p27^{Kip1} and Sox2 and taken confocal pictures to better show co-localization of p27^{Kip1} and Sox2 in the marginal zone of the mutant pituitaries (Figure 9, J-L). We detect expression of p27^{Kip1} in the marginal zone of the mutant BrafV600E pituitaries (Figure 9, arrows in J) whilst very few p27^{Kip1} positive cells are seen in the Wt marginal zone (Figure 9, G). Confocal images of the marginal zone (Figure 9, I’&L”) reveal that in the mutant pituitaries the cell cycle inhibitor p27^{Kip1} is co-expressed with Sox2+ve cells (Figure 9, arrowheads in L”) whilst no co-localisation of p27^{Kip1} was seen in Wt (Figure 9, arrows in I’).

These data, together with co-expression of p57^{Kip2} and Sox2 in the marginal zone of the mutant pituitaries (Figure 9, A-F), the increase in cell cycle inhibitors shown by q-PCR and *in situ* hybridisation (Figure 8), and the upregulation of cell cycle inhibitors by the Sox2+ve mutant cells *in vitro* (Figure 10) indicate that persistent activation of MAPK by expression of BrafV600E leads to increased expression of cell cycle inhibitors, concomitant with cell growth arrest in the mutant pituitaries over time.

“Fig7 and 12: the counting protocols are not detailed enough”

The counting has been added in material and methods.

“Fig10, SupFig9 and 15: the stage at which experiments were performed does not appear in the legend”

We have introduced the developmental stage in all corresponding figure legends.

SupFg13G: The proliferative capacity of the mutant pituitaries is said to significantly decrease but this is not clear on the figure.

We made a mistake in the figure legend with the wrong statement “*the proliferation capacity of the mutant pituitaries significantly declines compared to Wt*”. This sentence has now been removed (see Supplementary Figure legend 21). We have left “the proliferation rate is **comparable** between genotypes at E15.5 and E18.5, with **a trend to less proliferation at E18.5**” and we have added “although not statically significant”. Also, in the main text it says line 329-330 “from E15.5 no significant differences were noted”. We have also replaced the old pHH3 immunohistochemistry (Supplementary Figure 21. F) that had some background staining in the posterior lobe, with a better representative section.

“Ref36 does not correlate with expected citation”

We have removed this reference from the main text.

Typos: lines 407 412 415 424.

These have been corrected.

Reviewer #2 (Remarks to the Author):

Specific comments:

1. Clinical description of the patients in the main text?

The authors are unsure if the reviewer would like us to move the clinical description of the patients into the supplementary data, or to shorten the clinical description in the main text?

2. Mitotic index described in the text; it is not entirely clear if this index is increased, decreased or unaltered

This has been rephrased in the main text and materials and methods. Mitotic Index (MI) refers to the proportion of pHH3+ve cells expressed as % (MI= pHH3+ve cells/total number of cells X 100). Activation of MAPK causes a transient increase in proliferation at early stages of development (seen by increased MI); however later on in development, the MI decreases as a consequence of persistent activation of the MAPK pathway which leads to senescence, cell growth arrest, and apoptosis leading to pituitary hypoplasia.

3. Some spelling errors: whether (407), distinct (415), die in the (432), apoptosis (444)

These have been changed.